# Cervicovaginal microbiota and local immune response modulate the risk of spontaneous preterm delivery

Michal A. Elovitz[1], Pawel Gajer[2], Valerie Riis[1], Amy G. Brown[1], Michael S. Humphrys[2], Johanna B. Holm[2] & Jacques Ravel [2]

Failure to predict and understand the causes of preterm birth, the leading cause of neonatal morbidity and mortality, have limited effective interventions and therapeutics. From a cohort of 2000 pregnant women, we performed a nested case control study on 107 well-phenotyped cases of spontaneous preterm birth (sPTB) and 432 women delivering at term. Using innovative Bayesian modeling of cervicovaginal microbiota, seven bacterial taxa were significantly associated with increased risk of sPTB, with a stronger effect in African American women. However, higher vaginal levels of β-defensin-2 lowered the risk of sPTB associated with cervicovaginal microbiota in an ethnicity-dependent manner. Surprisingly, even in *Lactobacillus* spp. dominated cervicovaginal microbiota, low β-defensin-2 was associated with increased risk of sPTB. These findings hold promise for diagnostics to accurately identify women at risk for sPTB early in pregnancy. Therapeutic strategies could include immune modulators and microbiome-based therapeutics to reduce this significant health burden.

[1] Maternal and Child Health Research Center, Department of Obstetrics and Gynecology, Perelman School of Medicine, University of Pennsylvania, Philadelphia, PA 19104, USA. [2] Institute for Genome Sciences and Department of Microbiology and Immunology, University of Maryland School of Medicine, Baltimore, MD 21201, USA. Correspondence and requests for materials should be addressed to M.A.E. (email: melovitz@obgyn.upenn.edu) or to J.R. (email: jravel@som.umaryland.edu)

Preterm birth (PTB) (defined as birth before 37 completed weeks of gestation) is the leading cause of death in neonates and children under the age of 5[1,2]. Every year worldwide, 1.1 million babies die from consequences of prematurity. PTB occurs in one out of every 10 pregnant women in the United States and over 65–75% of all PTBs are spontaneous with the idiopathic onset of cervical change, uterine contractility and/or rupture of fetal membranes, while the remaining PTBs are medically indicated for reasons such as preeclampsia or fetal distress[3]. The economic burden of preterm birth is staggering, with an estimated cost of $26 billion per year in the United States alone[4,5]. While there is known racial disparity in spontaneous preterm birth (sPTB) with African-American women having significantly higher rates than non-African American women, factors that underpin this disparity remain elusive[6]. While there are medical, societal, and economic costs to the actual PTB, the larger cost to our society stems from the need for long-term care for these preterm infants[7]. Ex-preterm children are at increased risk for a spectrum of neurobehavioral disorders—ranging from cognitive deficits to cerebral palsy to neurobehavioral abnormalities including autism[8–10]. A failure to understand the causes of PTB have limited effective interventions and therapeutics.

The interaction between microbial communities and their host, in many biological niches, has been found to be mechanistically involved in health and disease pathogenesis[11–17]. To date, there have been a few studies that have examined the relationship between cervicovaginal microbial communities and sPTB[18–22]. Definitive conclusions from these studies are difficult to establish as phenotyping of sPTB is heterogenous, the number of sPTB cases is significantly limited and methodology is variable.

Here, to overcome sample size limitations, misclassification of cases and methodological differences, we conducted a study involving a prospective cohort of 2000 women with singleton pregnancies called Motherhood & Microbiome (M&M) and tested associations of cervicovaginal microbial communities and local immunological features with sPTB. A nested 1:4 case control study on 107 well-phenotyped cases of sPTB and 432 women delivering at term as control, and frequency matched for race, was performed after enrollment was completed and all delivery adjudicated (Table 1 and Supplementary Table 1). The population studied was mostly African American (AA) (74.5%) with a mean maternal age of around 28 years old. Characteristics associated with sPTB were statistically different, including history of sPTB or second trimester loss, cervical length, cerclage, and vaginal bleeding in the second trimester, while no other demographic, behaviors or clinical factors were different between the cases and controls at baseline and at each visit (Table 1 and Supplementary Table 1). Cervicovaginal samples and anthropometric measurements were prospectively collected during three clinical visits between 16–20 (visit 1), 20–24 (visit 2), and 24–28 (visit 3) weeks of gestation. The cervicovaginal microbiota was characterized and immunological profiles established.

## Results

**Cervicovaginal community state types sPTB.** We identified six major cervicovaginal community state types (CSTs)[23], of which four were predominated by either *Lactobacillus crispatus* (CST I), *Lactobacillus gasseri* (CST II), *Lactobacillus iners* (CST III) or *Lactobacillus jensenii* (CST V), and two (CST IV-A and CST IV-B) comprised a wide array of strict and facultative bacterial anaerobes, where CST IV-A was characterized with the higher abundance of BVAB1. The frequency of CSTs (Supplementary Table 2) was significantly different in AA and non-African American (non-AA) women (Fig. 1a). At visit 1, 20% and 45% of AA women were in CST I or CST IVA/IVB, as compared to 50%

| Table 1 M&M participants demographics and characteristics | | | |
|---|---|---|---|
| | Term (n = 432) | sPTB (n = 107) | *p*-value** |
| Race | | | 0.99 |
| African American | 322 (74.5) | 80 (74.8) | |
| White | 92 (21.3) | 23 (21.5) | |
| Other | 18 (4.2) | 4 (3.7) | |
| Maternal age in years, mean (SD) | 28 (6) | 29 (6) | 0.33 |
| Marital status | | | 0.24 |
| Single | 309 (71.5) | 70 (65.4) | |
| Married | 123 (28.5) | 37 (34.6) | |
| Insurance | | | 0.2 |
| Private | 212 (49.1) | 45 (42.1) | |
| Medicaid | 220 (50.9) | 62 (57.9) | |
| Nulliparous | 188 (43.5) | 38 (35.5) | 0.15 |
| Gestational diabetes | 17 (3.9) | 8 (7.5) | 0.12 |
| Pre-gestational diabetes | 9 (2.1) | 3 (2.8) | 0.71 |
| Chronic hypertension | 21 (4.9) | 8 (7.5) | 0.34 |
| History of sPTB/2nd trimester loss | 49 (11.3) | 43 (40.2) | <0.001 |
| Cervical length screening performed at Level II Ultrasound[a] | 368 (85.2) | 71 (66.4) | <0.001 |
| Cervical length at Level II (mm), median (IQR) | 34 (31–38) | 29 (20–33) | <0.001 |
| Cerclage | 8 (1.8) | 15 (14.0) | <0.001 |
| Vaginal bleeding—first trimester | 62 (14.4) | 16 (14.9) | 0.88 |
| Vaginal bleeding—second trimester | 25 (5.8) | 23 (21.9) | <0.001 |
| Vaginal bleeding—third trimester | 48 (11.1) | 18 (19.3) | 0.04 |
| Gestational age at delivery in weeks, median (IQR) | 39 (39–40) | 33 (25–35) | <0.001 |

**p-values estimated using *t*-test
[a]women with prior PTB are screened prior to level II ultrasound (16–22 weeks) and not all women undergo screening again during the specific level II ultrasound (19–21 weeks) which is performed to assess fetal anatomical structures

and ~15% of non-AA women, respectively. These differences persisted at visit 2 and visit 3 (Supplementary Fig. 1a). The frequency of CST III was higher in AA women at visit 3. Conversely, CST V was consistently lower in AA than in non-AA women throughout the study. When considering all women in the study at visit 1, CST IV-B was statistically significantly higher in the sPTB group than in the term group (Fig. 1b). Interestingly, in non-AA women a statistically significant association between CST IV-B and sPTB was observed when all visits were considered, at visit 1 and 2 (Fig. 1b and Supplementary Fig. 1b). This association was absent in African American women (Fig. 1b) at all study visits. In non-AA women, higher frequency of CST V was associated with term pregnancy, while the opposite was shown in AA women, however, the frequency of this CST was low in the cohort. These findings point to different contributions of the cervicovaginal microbiota to sPTB in AA and non-AA pregnant women, a hypothesis we thus addressed and considered in all analyses.

**Bacterial taxa associated with sPTB.** Because within a CST the specific relative abundance or presence of certain bacterial taxa can vary, we explored the dependence of the risk of sPTB on the relative abundance of specific bacterial taxa (Supplementary Data 1 and 2) using a Bayesian logistic regression nonparametric adaptive spline model while accounting for measurement errors obtained when sampling low relative abundance taxa, and

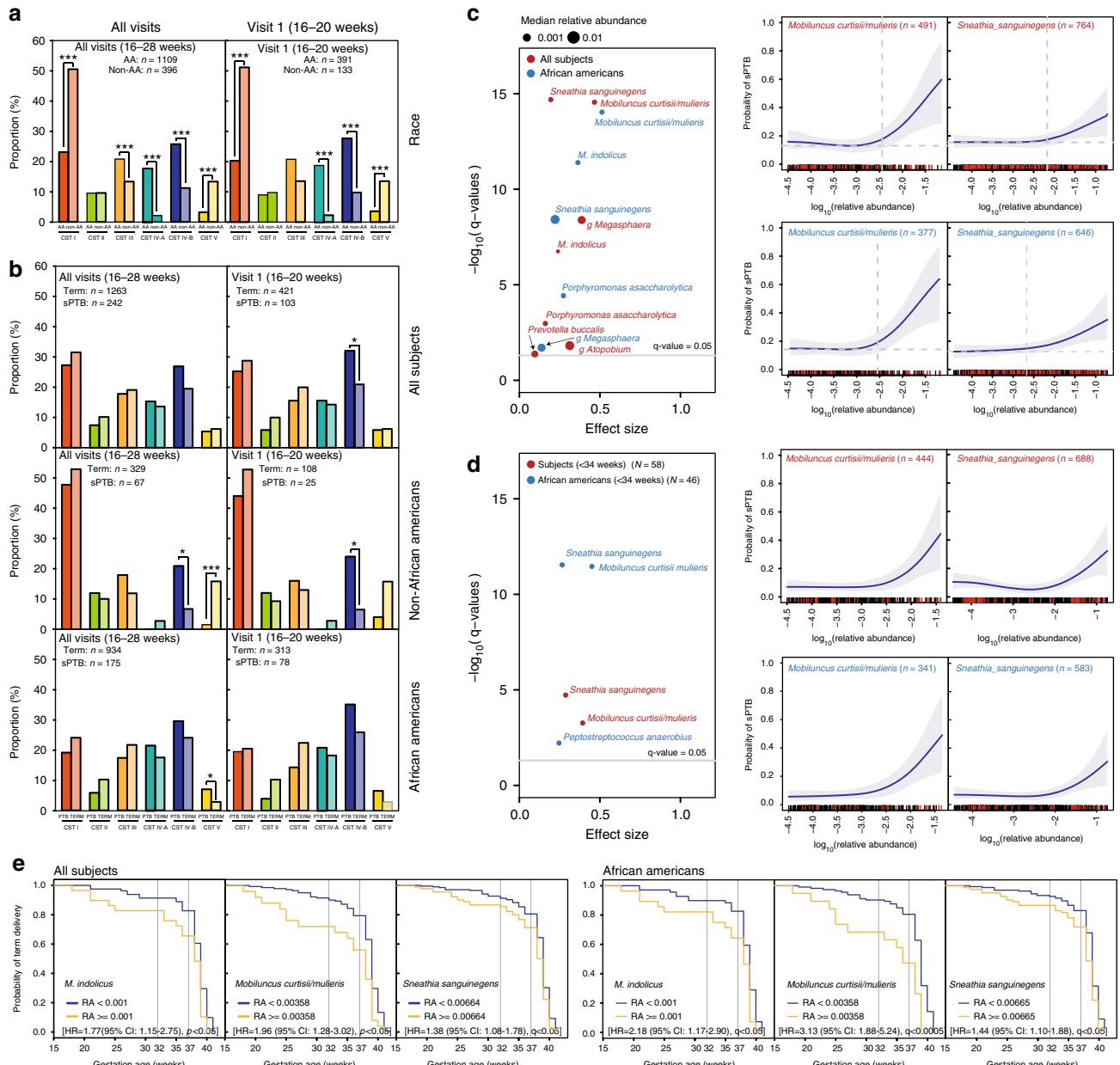

**Fig. 1** Vaginal microbiota composition and structure and risk of sPTB. **a** Frequency of each CST in non-AA and AA pregnant women when considering samples collected at all three visits or visit 1 (16–20 weeks of gestation). **b** Frequency of each CST stratified by outcomes (sPTB vs term) in all subjects, non-AA and AA when considering samples collected at all three visits or visit 1 alone. *n* represents the number of samples included in the analysis. For (**a**) and (**b**) *p*-values were estimated using mixed effects Poisson regression models (all visits) or ordinary logistic regression models (single visit) (**c**) Volcano plot for seven bacterial taxa statistically significantly associated with increased risk of sPTB in all subjects (red) and AA (blue) showing the effect size on the *x*-axis and the strength of the association on the *y*-axis. The gray horizontal lines indicate *q*-value of 0.05. The median relative abundance of each phylotype is indicated by the size of the point. Dependence of the risk of sPTB (defined as <37 weeks of gestation) on the log$_{10}$ relative abundance of *M. curtisii/mulieris* and *S. sanguinegens* in all subjects and AA is shown on the right. Effect size is the difference between the lowest and highest probability of sPTB. Greyed area indicates 95% credible region. Dotted line corresponds to the significant risk of sPTB threshold values (taxa log$_{10}$ relative abundance above which the risk is significant different from baseline). *n* represents the number of samples in which the bacterial taxon was detected and included in the analysis. (**d**) is the same as (**c**) but with sPTB defined as birth at <34 weeks of gestation. *N* represents the number of subjects in each group. Statistically significant taxa were identified using a Bayesian logistic regression nonparametric adaptive spline models. **e** Kaplan–Meier survival plot for *M. indolicus*, *M. curtisii/mulieris* and *S. sanguinegens* in all and AA women who harbor these bacterial taxa at relative abundance (RA) below (blue) or above (orange) the threshold values above which the risk of sPTB is significant different from baseline. *p*-values estimated using Cox proportional hazard regression models using *coxph()* routine of the *survival* R package. Statistical significance is shown as **p*-value < 0.01, ***p*-value < 0.001, and ****p*-value < 0.0001

modeled using a dataset generated from 14 sequencing positive control samples (Supplementary Data 3). When considering all subjects and all visits, we identified seven bacterial taxa statistically significantly associated with increased risk of sPTB (Fig. 1c). The strongest association was observed with *Mobiluncus curtsii/mulieris* and *Sneathia sanguinegens*, with *M. curtsii/mulieris*, g *Atopobium*, and g *Megasphaera* exerting the most profound effect on the risk of sPTB, with effect sizes of 0.33 for g *Atopobium*, 0.40 for g *Megasphaera*, and 0.46 for *M. curtsii/mulieris* (*q*-value < 0.0001) (Fig. 1c, Supplementary Fig. 2a and Supplementary Table 3). We established for each taxon a relative abundance threshold above which the risk of sPTB is significantly different from baseline (Fig. 1c). When all subjects and samples collected at visit 1 were considered, *M. curtsii/mulieris* was the only bacterial taxa associated with increased risk of sPTB (Supplementary Fig. 2c). In AA women, the increased relative abundances of five of these bacterial taxa remained significantly associated with increased risk of sPTB (Fig. 1c and Supplementary Fig. 2b). *M. curtsii/mulieris* and *Mageebacillus indolicus* had the strongest association (*q*-value < 0.0001) and the largest effect sizes (0.53 and 0.36, respectively), with a rate of sPTB over 60 and 55% at the highest relative abundance of these two bacterial taxa (Fig. 1b and Supplementary Fig. 2b). The association with *M. curtsii/mulieris* remained when only samples collected at visit 1 were considered, but with a lower effect size of 0.27 (*q*-value = $2.03e^{-0.5}$) (Supplementary Fig. 2d and Supplementary Table 3). In non-AA women, *L. iners* and *Atopobium vaginae* were significantly associated with increased rates of sPTB, but with modest effect sizes of 0.15 and 0.11, respectively (Supplementary Table 3). When comparing nulliparous and multiparous women, g *Megasphaera*, g *Atopobium*, and *S. sanguinegens* were strongly associated with sPTB in nulliparity (effect sizes above 0.24) but not in multiparity where *M. curtsii/mulieris* alone showed significance with an effect size of 0.44. The identification of different features of the cervicovaginal microbiota and their contribution to the risk of sPTB in AA and non-AA, CST types, or parity will afford a stratification scheme of risk and improved clinical managements.

**Survival analysis**. A survival analysis for term delivery as a function of gestational age in all pregnant women with these seven bacterial taxa at relative abundances above or below their significant risk of sPTB threshold values (Fig. 1c and Supplementary Table 3) identified *M. curtsii/mulieris, M. indolicus*, and *S. sanguinegens* as significantly increasing the risk of sPTB with hazard ratios (HR) ranging from 1.38 to 1.96 (estimated using Cox proportional hazard regression models) (Fig. 1e). These analyses demonstrate strong associations between the relative abundance of these taxa and sPTB but notably with early sPTB occurring at <34 weeks (Fig. 1d). When limiting the analysis to AA pregnant women, HR increases ranging from 1.44 to 3.13 for *M. curtsii/mulieris* (Fig. 1c–e).

**Bacterial taxa absolute abundance and sPTB**. Because quantitative estimates of bacterial taxa have high translational potential to clinical settings, we explored their relationship with birth outcomes. We showed that the total bacterial burden was significantly lower after 24 weeks of gestation in women who went on to deliver preterm compared to those who delivered at term in whom bacterial burden remained high (Supplementary Fig. 3a), while no differences in Shannon diversity was observed over gestational age (Supplementary Fig. 4). Interestingly, higher absolute abundance of *M. curtsii/mulieris* remained significantly associated with the increased risk of sPTB when all visits were considered in all subjects and in AA alone, as well as in women

with cervicovaginal microbiota of CST IV-B types (Supplementary Fig. 3b and Supplementary Table 4).

**Risk of sPTB modulation by *Lactobacillus* spp**. It is believed that *Lactobacillus* spp. is generally associated with positive reproductive and health outcomes[24]. We thus explored a potential modulatory role for *Lactobacillus* spp. of the risk for sPTB associated with *M. curtsii/mulieris*. The risk associated with *M. curtsii/mulieris* is unchanged and remains high when *Lactobacillus* relative abundances are in the lowest tertile, while it is eliminated when *Lactobacillus* relative abundances are in second or third tertiles. This finding was evident in both AA women (Fig. 2a) and in all subjects (Supplementary Fig. 5a) at visit 1. This finding supports a beneficial role for *Lactobacillus* spp. even in the presence of bacterial taxa associated with a significant increased risk for sPTB.

**Risk of sPTB and β-defensin-2**. We sought to assess if local immune responses would further modulate this microbe-associated risk of sPTB and explored the interaction between cervicovaginal levels of β-defensin-2 and pregnancy outcomes in the M&M cohort. β-defensin-2 are host-derived anti-microbial peptides that, in the genital track, can be constitutively expressed or induced upon bacterial infections[25]. β-defensin-2 were measured at visit 1 (Supplementary Table 5). Comparison of β-defensin-2 levels by pregnancy outcomes shows that women who ultimately had a sPTB had lower levels of β-defensin-2 (Fig. 2b). This association remains in AA women, but not in non-AA women (Supplementary Fig. 6). More interestingly, non-AA women who delivered at term had lower levels of β-defensin-2 than AA women who delivered preterm. This association appears to be modulated by the structure of the cervicovaginal microbiota. Lower β-defensin-2 levels were observed in AA women who had a sPTB and harbor a cervicovaginal microbiota of CST III, CST IV-A, and CST V (Fig. 2b), while in all subjects, lower levels of β-defensin-2 were found in women who delivered preterm and harbor CST I, CST II, and CST IV-A (Supplementary Fig. 5b). In AA women, when stratifying β-defensin-2 levels in quartiles, the risk of sPTB associated with the presence of g *Atopobium, M. curtsii/mulieris, S. sanguinegens*, and *M. indolicus* was increased in the first quartile and lowered in the fourth quartile (Fig. 2c), indicating that β-defensin-2 levels modulate the risk of sPTB associated with the presence of some of these bacterial taxa but not all (i.e., g *Megasphaera*).

**Interactions between risk modulators and sPTB**. These results led us to consider the possibility that a complex interaction exists between the modulators (β-defensin-2 levels and relative abundance of *Lactobacillus* spp.), *M. curtsii/mulieris*, CSTs and risk of sPTB (Fig. 3 and Supplementary Table 6). We showed that in AA women the risk of sPTB is high even in the presence of *Lactobacillus* spp. (including *L. crispatus* dominated CST) when β-defensin-2 levels are low (1st tertile) (*p* = 0.011), and that sPTB rates trended lower when β-defensin-2 levels are high (3rd tertile) and that whether the microbiota is dominated by *Lactobacillus* spp. (CST I, II, III or V) (3rd tertile) or comprising of strict and facultative anaerobes and low *Lactobacillus* spp. (first and second tertile). Most women with relative abundances of *M. curtsii/mulieris* above sPTB risk threshold were associated with the second and third tertiles of both β-defensin-2 and *Lactobacillus* spp. (Fig. 3) Interestingly, β-defensin-2 levels in the second tertile did not lower the risk of sPTB with microbiota of CST IV-A and IV-B, but high relative abundance of *Lactobacillus* spp. trended towards modification of the risk (Fig. 3 and Supplementary Table 6). Similar results were observed when including all

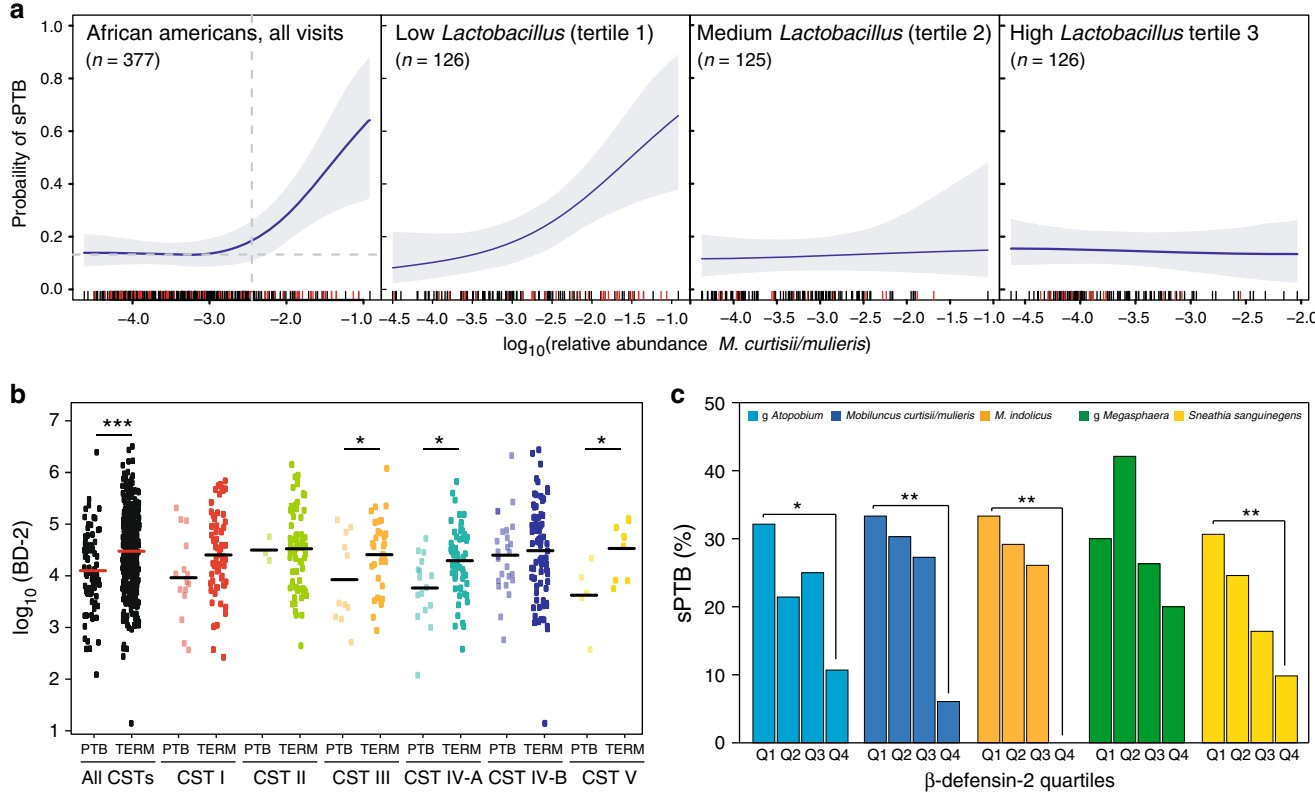

**Fig. 2** β-defensin-2 and microbiota modulate the risk of spontaneous preterm delivery. **a** Modulation of the risk of sPTB by relative abundance of *M. curtisii/mulieris* stratified by *Lactobacillus* spp. relative abundance tertiles within AA when all visits are considered. *n* represents the number of samples where *M. curtisii/mulieris* was detected. **b** log$_{10}$ β-defensin-2 abundances at visit 1 in AA women stratified by pregnancy outcomes and vaginal community state types. *p*-values estimated using a *t*-test. **c** At visit 1 in AA women, the risk of sPTB associated with the relative abundance of five bacterial taxa is modulated by the abundance of β-defensin-2. *p*-values were estimated using a Bayesian 2-proportions binomial model with uniform prior implemented in *rstan* R package. Statistical significance is shown as *$p$-value < 0.01, **$p$-value < 0.001 and ***$p$-value < 0.0001

subjects in the analysis (Supplementary Fig. 7 and Supplementary Table 6) and when combining all samples with high β-defensin-2 (3rd tertile) and all tertiles of *Lactobacillus* spp. (Supplementary Table 6). These findings start to unravel the complex interactions that exist in the cervicovaginal environments between immune factors, the microbiota and the risk to sPTB. While this study is well-powered and the associations statistically significant, a larger number of cases would afford greater statistical power to explore additional factors that are thought to contribute to increasing or reducing the risk of sPTB.

## Discussion

The study has major implications to the clinical management of pregnant women, as it identified specific signatures combining both immune and microbial factors associated with sPTB. The latter include both risk factors and risk modulators. The findings address a long-held belief that not having *Lactobacillus* spp.-dominated cervicovaginal microbiota is strongly associated with adverse pregnancy outcomes. As shown here and in previous studies[20,22] in pregnancy and non-pregnancy, a larger proportion of AA women compared to non-AA do not have *Lactobacillus* spp. in high relative abundance in the cervicovaginal microbiota. While the rate of sPTB is higher in AA women, the difference cannot be explained by the lack of *Lactobacillus* spp. as many AA women deliver at term despite lacking *Lactobacillus* spp. Conversely, harboring a *Lactobacillus* spp. does not guarantee a term birth in both AA and non-AA women. Further, different causes of sPTB appear to drive risk in AA and non-AA women. Here we

show that immune factors, such as β-defensin-2, can modulate the risk associated with the lack of *Lactobacillus* spp., but are also critical even when *Lactobacillus* spp. are in high relative abundance. However, the reasons why some women have high or low β-defensin-2 levels is unknown. Differences in local cervicovaginal immune profiles in pregnancy has been poorly documented, especially as it relates to sPTB, and mixed results have been reported between cervicovaginal microbiota and β-defensins[26,27]. Genetic variants or copy number[28] in β-defensin-2 genes, or changes in response to bacterial exposures, or a combination of both could explain such variations, including the identified ethnic differences, but environmental exposure factors cannot be excluded to play a role in the shaping of the cervicovaginal environment and the risk to sPTB. Alternatively, one can speculate that early exposure to non-*Lactobacillus* spp. cervicovaginal microbiota at birth could lead to tolerance to these predominantly anaerobic bacteria later in life resulting in specific immune profiles. Nonetheless, the profiles identified in this study could serve as diagnostic signatures with important clinical relevance. However, the clinical significance of any differences or microbial immune correlations cannot be fully interpreted until these biomarkers are validated in another clinical trial outside of the case-control study.

Women with a sPTB present with contractions, cervical dilatation and/or preterm premature rupture of membranes. As women can present with several of these symptoms, clear phenotyping by preterm rupture of membranes or cervical dilation is problematic. Therefore, for clinical studies, these various clinical presentations are collectively characterized as sPTB. Current

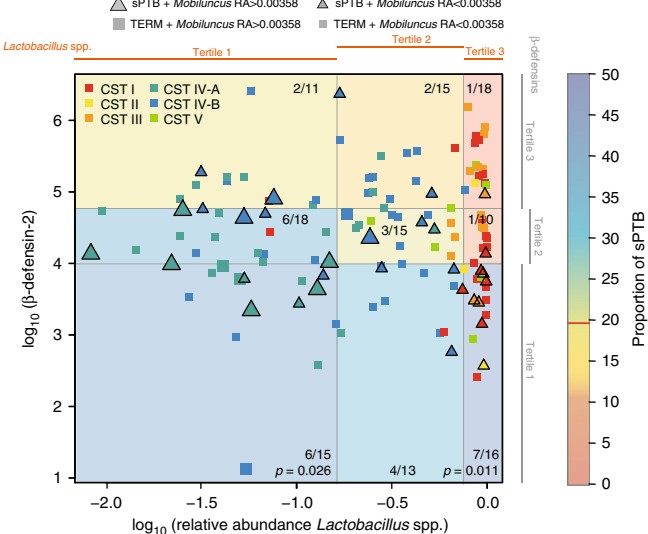

**Fig. 3** Interactions between β-defensin-2, *Lactobacillus* spp. relative abundance, *M. curtisii/mulieris* relative abundance, vaginal community state types and pregnancy outcomes in AA women at visit 1. Each woman who went on to deliver preterm is represented by a triangle, while those who delivered at term by a square. Large and small triangles or squares are colored by CSTs and indicate relative abundances (RA) of *M. curtisii/mulieris* above and below its threshold value, respectively, as defined in Fig. 1. β-defensin-2 concentrations and *Lactobacillus* spp. relative abundances were stratified into tertiles. The color of each quadrant indicates the proportion of sPTBs (number of sPTB/total births). *p*-values were estimated using Bayesian binomial models using uniform prior for two proportions implemented in *rstan* R package

clinical approaches to identify and reduce the risk of sPTB include screening for short cervical length in all women and treating women with a prior sPTB with 17-alpha hydroxyprogesterone caproate (17OHPC). These strategies are limited. Only 9% of all sPTB have an antecedent short cervix in mid-pregnancy[29]. While vaginal progesterone has been shown to confer some protection against sPTB in women with a short cervix, considering the small percent of women who have a short cervix this strategy will not significantly impact the rate of sPTB[29,30]. A prior sPTB is the largest known risk factor for sPTB. Based on a clinical trial, 17OHPC is recommended for all women with a prior sPTB[31,32]. However, only a small percent of the total number of women having a sPTB had a prior sPTB, and the therapy is effective in only 1/3 of those who receive it[31]. Consequently, the current standards of care to screen for a short cervix or/and or targeting women with a prior sPTB will not significantly decrease the public health burden from sPTB[33]. Studies such as this one will lead to novel sPTB prevention strategies based on appropriate risk stratification of pregnant women and informed personal counseling. Most importantly, this work will lead to innovative therapeutic opportunities to prevent sPTB including combination of microbiome-based therapeutics and immune modulators earlier in pregnancy.

## Methods
**Study design and sample collection**. A prospective cohort study of 2000 women with singleton pregnancies was utilized M&M, of which biospecimens were obtained for 1943 participants and delivery information was available for 1920 from whom the case/controls were derived. All participants provided written informed consent and the study was approved by the Institutional Review Board at the University of Pennsylvania (IRB #818914) and the University of Maryland School of Medicine (HP-00045398). Women were enrolled between December

2013 and February 2017. Enrollment had to occur prior to 20 weeks of pregnancy. Cervicovaginal specimens were self-collected by the participant or collected by a research coordinator if a clinical exam was indicated at three different prenatal visits: 16–20 weeks (visit 1), 20–24 weeks (visit 2), and 24–28 weeks (visit 3). Prior to each visit, subjects were asked regarding antibiotic use, sexual activity, and tobacco use in the 4 weeks prior to the visit. At each visit, a set of cervicovaginal swabs was obtained. These included an ESwabs (COPAN) stored in 1 ml of Amies Transport Medium and a Dacron swab stored without buffer. All samples were immediately frozen at −80 °C until processing. Subjects were followed to delivery. All delivery outcomes were recorded. Cases of PTB were adjudicated by the PI to determine if the cases were medically indicated PTBs or spontaneous. PTB was considered spontaneous when a woman presented with either cervical dilation and/or premature rupture of membranes and delivered prior to 37 weeks. Women delivered for maternal indications (e.g., preeclampsia) or fetal indication (e.g., fetal growth restriction) were not considered sPTB. Women were eligible for the study if they had a singleton pregnancy and presented prior to 20 weeks of gestation. Women were excluded if they met any of the following criteria: (1) major fetal anomaly, (2) HIV positive status, (3) history of organ transplant, and (4) chronic steroid use (>20 mg per day for more than 30 days at the time of first study visit). To assess the cervicovaginal microbial communities, a nested case-control selection of subjects was performed. Among the 1943 women who completed the study, all women who had a sPTB were selected. Controls were frequency matched by self-reported race to the cases.

**Cervicovaginal 16S rRNA gene amplification and sequencing**. Cervicovaginal ESwabs were thawed on ice, and 300 µl of Amies transport medium containing vaginal secretion were processed using the MoBio PowerMag Microbiome DNA/RNA kit (now MagAttract PowerMicrobiome DNA/RNA kit, Qiagen) automated on a Hamilton Microlab STAR robotic platform after a bead-beating step on a Qiagen TissueLyser II (20 Hz for 20 min) in 96 deep well plate. Amplification of the V3–V4 regions of the 16S rRNA gene was performed using a two step-PCR in which the sample specific barcode is added during the second PCR. The first PCR used the short 16S rRNA gene specific primers 319F (ACACTGACGACATGGT TCTACA[0–7]**ACTCCTRCGGGAGGCAGCAG**) and 806R (TACGGTAGCAG AGACTTGGTCT[0–7]**GGACTACHVGGGTWTCTAAT**) where the underlined sequence is the Illumina sequencing primer sequence and [0–7] indicate the presence of an heterogeneous pad sequence to improve sequencing quality[34], for a total of 20 cycles. This first step was followed by 10 cycles with primers H1 (AATGATACGGCGACCACCGAGATCTACACNNNNNNNNNACACTGACGAC ATGGTTCTACA) and H2 (CAAGCAGAAGACGGCATACGAGATNNNNNNNN NTACGGTAGCAGAGACTTGGTCT) where NNNNNN indicates a sample specific barcode sequence and the underlined sequence corresponds to the Illumina sequencing primer for priming to the first step amplicon. This second step extends the amplicon with the Illumina required adaptor sequences and the sample specific dual barcode system[34]. Amplicons were visualized on a 2% agarose gel, quantified, pooled in equimolar concentration, and purified prior to loading on an Illumina HiSeq 2500 (San Diego, CA, USA) modified to generate 300 bp paired-end reads[35]. Extraction and PCR negative controls were processed in parallel. Additionally, a positive control composed of a mixture of 20 vaginal biological specimens of known composition combined into one tube was processed and sequenced in parallel on each of 14 pools of study samples as per the laboratory standard protocol (Supplementary Data 3).

**Total 16S rRNA gene copy number measurement**. The total number of 16S rRNA gene copy in each DNA sample was measured using the TaqMan® Bact-Quant assay targeting the V3–V4 regions of the gene[36]. Amplification primers were FOR (CCTACGGGDGGCWGCA) and REV (GGACTACHVGGGTMTCT AATC), and TaqMan® probe was 6FAM (fluorescin)-CAGCAGCCGCGGTA-M GBNFQ (minor groove binder non-fluorescent quencher). The total number of 16S rRNA gene copies was expressed as copy per swab and is used as an estimate of bacterial load (total count of bacterial cells present in a sample). An estimate of absolute abundance of each taxon was calculated for each sample by multiplying the total 16S rRNA gene copies obtained by qPCR and the relative abundance of that taxa obtained by 16S rRNA gene sequencing.

**Bioinformatics methods for microbiota analysis**. The sequences were de-multiplexed using the dual-barcode strategy, a mapping file linking barcode to samples and split_libraries.py, a QIIME-dependent script[37]. The resulting forward and reverse fastq files were split by sample using the QIIME-dependent script split_sequence_file_on_sample_ids.py, and primer sequences were removed using TagCleaner (version 0.16)[38]. Further processing followed the DADA2 Workflow for Big Data and dada2 (v. 1.5.2) (https://benjjneb.github.io/dada2/bigdata.html,[39]). Forward and reverse reads were each trimmed using lengths of 255 and 225 bp, respectively, and filtered to contain no ambiguous bases, minimum quality score of 2, and required to contain less than two expected errors based on their quality score. The relationship between quality scores and error rates were estimated for both sequencing runs to reduce batch effects arising from run-to-run variability. Reads were assembled and chimeras for the combined runs removed as per dada2 protocol.

Taxonomy was assigned to each amplicon sequence variant (ASV) generated by dada2 using the PECAN (version 1.0), a rapid per sequence classifier (http://ravel-lab.org/pecan). Read counts for ASVs assigned to the same taxonomy were summed for each sample. Phylotypes previously identified as potential reagent contaminants[40], that included *Pseudomonas veroni*, *Achromobacter xylosoxidans*, g *Halomonas*, g *Micrococcus*, g *Heliorestis*, and *Arthrobacter cumminsii* were excluded from the dataset as they could be observed in some negative control samples. Community state types (CST) were assigned to each sample using hierarchical clustering with Jensen-Shannon divergence and Ward linkage[41,42]. CST I is predominated with *L. crispatus*, CST II with *L. gasseri*, CST III with *L. iners*, CST IV was defined as lacking *Lactobacillus* spp. and comprising a diverse set of strict and facultative anaerobes, and further split into CST IV-A and CST IV-B, while CST V is predominated with *L. jensenii*.

Tables including total sequence counts for or relative abundances of all taxa and for each sample with more than 1000 sequences were generated and used for statistical analyses (Supplementary Data 1 and 2). A total of 60,599,360 sequences were generated from 1505 samples (540 subjects) for a mean of 40,238 sequences per sample (range 1313–98,688). There were 242 samples from 107 subjects that had a spontaneous premature delivery (22, 35 and 50 subjects with 1, 2, and 3 samples, respectively) and 1263 samples from 432 subjects that delivered at term (0, 33 and 399 subjects with 1, 2, and 3 samples, respectively).

**β-defensin-2 measurements**. Dacron swabs (Starplex, ThermoFisher) were used to collect cervical vaginal fluid at each gestational time point. Materials on the swabs were eluted in sterile PBS with a protease inhibitor cocktail (Complete Mini) for 5 min to release the soluble proteins. The eluate solutions were then analyzed for the presence of human β-defensin-2 (BD-2) (Phoenix Pharmaceutical); minimum detectable concentration = 15.6 pg/ml; inter-assay variation <15%; intra-assay variation <10% via ELISA according to manufacturer's instructions. Individual samples were diluted to meet the range of measurement on the assay's standard curve. Dilution factors were then applied to calculate precise β-defensin-2 levels.

**Statistical analyses**. Taxa were filtered before analysis if observed at frequencies of $10^{-5}$ study-wide. All scripts used in statistical analyses are available in Github at https://github.com/ravel-lab/M_and_M. Mixed effects Poisson regression models were used to assess difference in the frequency of CSTs in women who delivered at term and those who delivered preterm, stratified by race or birth outcome (term vs sPTB). Ordinary logistic regression models were used for assessing differences in CST frequencies at single visits stratifying by race and birth outcome (term vs sPTB).

To evaluate the dependence of community diversity or bacterial absolute abundance ($\log_{10}$ 16S rRNA gene copy number) on gestational age, we applied ordinary least squares model, in which samples were assumed to be independent, with thin plate splines as implemented in the *rms* R package (https://cran.r-project.org/web/packages/rms/rms.pdf). Akaike Information Criterion (AIC) was used to select the number of knots in the splines. In both cases splines with three knots had the smallest AIC value and were selected as the best fit model.

Due to the 1:4 case-control design, the evaluation of the dependence of the risk of spontaneous preterm delivery on bacterial taxa relative abundance was performed on phylotypes that were present in more than 25% of samples study-wide. In order to overcome challenges associated with the bimodal distributions of taxa $\log_{10}$ relative abundances, we evaluated how the risk of sPTB depended on the $\log_{10}$ relative abundances of taxa that were present in at least 25% of all samples. In this analysis, we only included samples in which a given bacterial taxon was detected, as no significant difference in the proportions of samples from the sPTB or term groups were found when only considering samples in which a given bacterial taxa was not detected. Because this dependence a priori can have a complicated shape, a Bayesian logistic regression nonparametric adaptive spline model was used (adapted from the *spmrf* R package[43]) as the adaptive splines implemented in the *spmrf* package have superior behavior at the extremities of the independent variable value. In this model, all samples were assumed to be independent. Further, microbiota analyses suffer from low reproducibility when taxa are present at low relative abundance in a sample. In order to account for these measurement errors, we leveraged the sequence data of a sample comprising of several vaginal swabs that were mixed and sequenced on most plates of 96 samples processed for this study (Supplementary Data 3). Bayesian logistic regression nonparametric adaptive spline models were run multiple times ($n = 5$). Specifically, for a given taxa $\log_{10}$ relative abundance, a new $\log_{10}$ relative abundance for that taxon is drawn from the known normal distribution of $\log_{10}$ relative abundance values generated from the positive controls dataset for that $\log_{10}$ relative abundance. The final spline dependence of the risk of sPTB on each taxa $\log_{10}$ relative abundance was estimated as the average of those five random sample models. Because the variance around a given $\log_{10}$ relative abundance in the positive controls dataset is larger for low relative abundance taxa, the confidence of detected an effect on the risk of sPTB for a taxon at low abundance is lower, thus the spline trends toward baseline risk of sPTB, which is the proportion of sPTB cases in the samples harboring that taxa.

The *spmrf* spline models accounting for measurement errors were fitted to the entire dataset, and on subsets that included all samples stratified by race (African

American or non-African American), by race and visit (African American or non-African American women at visit 1), as well as by nulliparity status, and nulliparity status limited to visit 1. The significant risk of sPTB threshold values (taxa $\log_{10}$ relative abundance above which the risk is significant different from baseline) were estimated as the intersection of the risk of sPTB spline estimate, $f$, and horizontal line $y = \min(f) + 0.1*gEff$, where gEff is the effect size defined as the difference between the maximal and minimal risk of sPTB when risk of sPTB was positively associated with relative abundances, or negative of the difference when an inverse association was observed. Bacterial species with effect size of <10% were excluded. Statistical significance was set a priori at the 0.05 level after adjusting for multiple comparisons using FDR.

The analysis of the dependence of the risk of sPTB on taxa $\log_{10}$ absolute abundances was performed using the same approach as described for relative abundance using Bayesian logistic regression nonparametric adaptive spline models with the exception that the independent variable was $\log_{10}$ absolute abundance instead of $\log_{10}$ relative abundance.

Comparisons of the mean $\log_{10}$ β-defensin-2 abundances between cases and controls in all samples as well as between cases and controls in different CSTs in AA women at visit 1, were performed using the *t*-test as log β-defensin-2 abundances within these groups follow normal distribution.

To evaluate the modulation of the risk for sPTB associated with bacterial taxa by β-defensin-2, comparisons of sPTB proportions within different quartiles of β-defensin-2 over samples where selected phylotypes were detected was performed using a Bayesian 2-proportions binomial model with uniform prior implemented in *rstan* R package.

Time to delivery in the presence of specific bacterial taxa was visualized with Kaplan–Meier curves generated using the *survfit()* routine from the survival R package and hazard ratios with the corresponding *p*-values estimated using Cox proportional hazard regression models using *coxph()* routine of the *survival* R package. In these models, bacterial taxa were included as time varying covariates.

We evaluated the interaction between β-defensin-2 tertiles, *Lactobacillus* spp. tertiles, CSTs, *M. curtsii/mulieris* and sPTB. The proportion of sPTB in each cell of the $3 \times 3$ table shown in Fig. 3 was compared to the proportion of sPTB (19.58%) at visit 1 defined by the study design, and for which β-defensin-2 measurements are available: 102 women who went on to deliver prematurely and 521 women total for which we have β-defensin-2 measurements at visit 1. The *p*-value for the null hypothesis that the proportion of sPTB in a cell is the same as the baseline proportion of sPTB was estimated using Bayesian binomial models using uniform prior for two proportions implemented in *rstan* R package. Similar approach was used to compared proportion sPTB in combined cells to the baseline rate baseline 19.58%.

Note: In the above analyses, when samples from all three visits are included, given the number of samples are decreasing at each visit due to preterm delivery events, we faced the issue of missing data. The models do not account for the non-randomness of the missing data.

## Code availability

All analysis scripts used in this manuscript are available at: https://github.com/ravel-lab/M_and_M.

## Data availability

The sequence data and the associated samples and subjects' metadata generated under this study are publicly available in the database of Genotypes and Phenotypes (dbGaP) under accession number phs001739.v1.p1. Processed data and limited metadata are available in supplementary materials.

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

## Acknowledgements

The authors would like to thank Elias McComb, Honqiu Wang and Li Fu for their valuable contributions to the 16S rRNA gene sequencing effort., Katheryne Downes for her help with coordinating the study metadata and Laura Anglim for performing the β-defensin-2 ELISA. This study was supported by the National Institute for Nursing Research of the National Institutes of Health under award number R01NR014784.

## Author contributions

M.A.E. and J.R. conceived the study. M.A.E. and V.R. led the clinical study and immunology data generation. M.S.H. and J.R. led the microbiome data generation. A.B. generated the immunology data. P.G., J.B.H., M.A.E., and J.R. performed the analyses. J.R. and M.A.E. wrote the manuscript.

## Additional information

**Competing interests:** M.A.E., P.G., and J.R. are inventors on a patent application (number PCT/US2018/012185) submitted by the Trustees of the University of Pennsylvania that covers compositions and methods for predicting risk of preterm birth. The authors declare no other competing interests.

