## [Peer Review File · Nature Communications]

Reviewers' comments:

Reviewer #1 (Remarks to the Author):

The authors present the results of their nested case control study designed to better understand the relationship between the cervicovaginal microbiome and immune response and preterm birth. The particular relevance of this topic to the broader audience should not be underestimated. While the authors correctly identify the short term sequelae of preterm birth they fail to mention the long term health risks associated with preterm birth for both the mother and fetus. Understanding the etiology, identifying markers of risk and developing therapeutic strategies could have substantial impact on health care delivery in the US.

The manuscript is well written and the results advance our understanding of potential mechanisms that result in preterm birth as well as opportunities for biomarker discovery and therapeutic interventions. There are several concerns that should be addressed:

1. The use of cervicovaginal community state types (CSTs) has been a common approach for evaluation of the vaginal microbiome but also recently challenged. The approach of categorizing data based on prior published work may limit discovery of new or novel findings.
2. The authors use the term "cervicovaginal" in vaginal self-collected samples. It is not clear how they can be certain that the cervix is part of the sampling. Cervical microbiome warrants separate evaluation.
3. Although not clearly stated, it is assumed that race is defined by patient self-report. This definition can be problematic and analysis specifically designed to evaluate for differences in racial groups may introduce bias.
4. Other variables that might impact the microbiome are not considered (vaginal douching, oral sex etc)
5. The authors present the results of spontaneous preterm birth but do not consider whether those spontaneous preterm births were a result of spontaneous preterm labor or preterm premature rupture of membranes.

Reviewer #2 (Remarks to the Author):

Spontaneous preterm birth (sPTB) proportionally affects African American (AA) women. Elovitz et al. describe a very large study that tries to unravel (sic!) the many factors that might play a role in this tragic statistic. In this study of nearly 2000 women (three quarters of which were AA), microbial (vaginal microbiome composition and absolute bacterial load) as well as immunological (beta-defensin-2) were investigated. The main conclusions are:

- * Lower levels of *Lactobacillus* spp. is associated with sPTB in non-AA women but not in AA women.
- * Several taxa, but in particular a high abundance of *Mobiluncus curtisii/mulieris*, were associated with an increased risk for sPTB
- * A decrease of bacterial absolute abundance after 24 weeks of gestation was found in women who delivered preterm.
- * High *Lactobacillus* levels reduced the probability of sPTB in the presence of high *Mobiluncus curtisii/mulieris*
- * AA women who had sPTB had lower levels of b-defensin-2. However, for nonAA women, this was the other way around.
- * in AA women the risk of sPTB is high even in the presence of *Lactobacillus* spp. when b-defensin-2 levels are low

These results illustrate the complex interaction between potential pathogens, beneficial *Lactobacillus*, immune factors, and ethnicity. Even within this very large cohort, patterns are not completely clear. However, this study is an enormous step forward in the search for factors that can be used to predict whether or not a pregnancy will be term or not.

The text is very well written, and the figures are mostly clear, albeit dense and sometimes hard to read. Some suggestions for textual and visual improvements are written below.

Introduction

1. Line 32: Every year: Please add if this number is worldwide or in the US
2. L33. It might be helpful to add what the other 25% of PTB are, since readers (including me) might assume that nearly all PTB are spontaneous.

Results

3. L84. It came as a surprise that only 14 positive controls were used to find taxa associated with PTB when 107 sPTB controls were available. Maybe I am not understanding this model correctly - is this similar to machine learning where a subset is used for training and the remaining samples for confirmation? The description in the Methods was way over my head but some reasoning for only using this small subset would be welcome. When all women were considered, the model found fewer taxa. How was this subset chosen, randomly?
4. L90 "for each taxa" - should be "for each taxon"

Methods.

5. L220. The wording is not clear here. I assume that samples were not "self collected by the research coordinator" - so rewording would help.
6. L238. "MoBio Microbiome kit" - please provide manufacturer. Is there a more specific name for this kit?
7. L254. "a set of 14 positive controls composed of a mixture of vaginal biological specimen combined into one tube" was not very clear. Were 14 samples pooled into one tube? Or were there 14 samples that each consisted of a mixture of specimens? How many specimens? Should "specimen" be "specimens" here? Please reword to make it more clear what was done.
8. L296. "Swabs were washed in sterile PBS" is not clear. Do the authors mean that the swab was soaked (not washed) in PBS, and the PBS then was analyzed? Or did they wash the swabs with PBS, then discard the PBS and analyze the swabs?
9. L306. "All scripts used in statistical analyses are available in the Supplemental data" - I could not find these in the materials shared for peer review.
10. L323. The description of the "Bayesian logistic regression nonparametric adaptive smoothing model" was beyond my expertise and I could not peer review this section. The method described here appears to be a modification of an R package but I would not have any idea how to perform this.
11. L375. Sequence data was not available for peer review.

Figures and tables

12. Figure 1. Some of the text is very small. The quality of the graph is excellent, so I can zoom in and read it, but it might be worth investigating if some text could be printed a bit larger.
13. Figure 1e. I assume RA stands for "relative abundance", but it would be helpful if that could be defined in the legend text.
14. Figure 2a. It took a while to correctly interpret the 3 Lactobacillus graphs, partially because the "tertile" term might not be familiar for all. Would it be more clear if the titles were "Low

Lactobacillus", "Medium Lactobacillus", "High Lactobacillus", with additional explanation about the tertiles in the text? The authors can ignore this if this is not scientifically correct.

15. Figure 2a. Since the X axes are the same for all 4 panels, but not clearly labeled, it might be more clear to call them "log₁₀ (Relative abundance of *Mobiluncus curtisii*/mulieris) - the title of the axis might only be shown once. Just a suggestion to make it more clear that the axes shown are the same for all 4 graphs.

16. Figure 2b. Assuming that the first 2 datasets shown here (in black) are the combined samples irrespective of CST, it might be more clear to say "all" or "combined" under the graph.

17. Figure 3. I compliment the authors for this figure, which conveys a lot of information in single graph, and was very interesting to explore. However, I was not sure how to interpret the legend shown at the top. Here the authors want to add the abundance of *Mobiluncus* (high or low) but it is very unclear. Is only the size (small or big) and shape (triangle/square) important or also the outline and shading color? Should I be looking for large and small squares and triangles, or for grey or black shapes? The smaller "square" on the right appears to be shaded with an outline and an X in the middle, and is also not really a square but a rectangle. It was also not clear why all triangles had an outline and none of the squares. What do the high-abundance *Mobiluncus* triangles and squares have in common; their shading? This might need some revision.

18. Figure S3. I was not sure how to interpret this graph. Is the X axis showing the timepoint during pregnancy at which the sample was taken? Or is the X axis showing the number of weeks at which the baby was born (probably not since both Term and sPTB are shown?) I suspect it is the first, but it would help to better word what is shown here.

Reviewer #3 (Remarks to the Author):

This paper evaluates the association between vaginal microbiome and an immune marker with spontaneous pre-term birth. This is a well written paper using data from a large, prospective case-control cohort. Major strengths include the longitudinal collection of samples, inclusion of negative and positive controls, estimation of total bacterial burden and the approach taken to classify taxa. The authors have included a large amount of relevant supplementary material and publicly posted their code and data to better enable reproducibility of results. I offer some suggestions for the improvement of the paper.

Major

1) A strength of the study is the longitudinal design. The logistic regression includes a random effect which is presumably a random subject effect to account for repeated measures. This appears to be the only analysis in which the repeated measures were accounted for in the model. Accounting for repeated measures in all of the analyses using data from all visits might be especially important since controls were more likely to have all three samples collected whereas the sPTB groups had a more even distribution of 1,2 or 3 samples collected. Although this concern is somewhat tempered by the inclusion of analyses for each visit separately. Regardless, the ability to evaluate changes over time within a woman are not capitalized in the current analyses and there is the potential for missing important trends over time, for example there are women who switch CSTs over time, might this reflect important changes in addition to simply noting their current CST?

2) The effect of missing data is not addressed, while it may be assumed that some of the sample are missing at random due to inadequate samples or simply missed visits, it is also likely that some of visit 3 samples in the sPTB group are missing not at random depending on when the sPTB event occurred. Given this is already a complex series of analyses, I think it would be helpful to at least acknowledge this issue and briefly discuss potential implications.

3) In the description of the survival analysis it is not clear if data from all visits were included and if so, were the taxa included as time varying covariates? Was there any weighting or a frailty term included to account for the matched case-control design?

Minor

1) What was the definition of sPTB?

2) Table S1 shows that BMI is stable across the 3 visits, is this baseline BMI or is it adjusted for expected weight gain during pregnancy?

3) Results 5th paragraph: what is the distribution of RA of *M. curtisii*/*mulieris* in each of the *Lactobacillus* tertiles? Given that the relative abundances are used, an increase in *Lactobacillus* must necessarily result in a decrease of something else, might this result simply be an artefact of compositional data?

4) The decrease in the total bacterial burden is an interesting finding. Is this decrease observed over time within women or could this be an artefact of missing visit 3 samples in women who had a PTB?

5) Is a difference of 0.3 logs for beta-defensin-2 clinically meaningful? With a large sample size, small effect sizes can be statistically significant but may not be clinically different. This is also true of the individual taxa selected, the median relative abundance is very small.

6) What is the reasoning behind removing samples with fewer than 1000 sequences, was the total bacterial burden low for these samples?

- 7) The methods indicate that SLPI was also measured but these results aren't presented.
- 8) The statistical analysis mentions that the logistic regression for sPTB controls for race and birth outcome. What is birth outcome and how does this differ from sPTB?
- 9) How were the 0's handled prior to the log transformation for the relative abundance?
- 10) Figure 3 & S7, it is difficult to distinguish points above and below the *M. curtsii/mulieris* cutoff.
- 11) Table 1: It seems odd to me that the controls would be screened more often for cervical length compared to women with sPTB, is this because these women were more likely to have a history of previous sPTB? Are there other factors that contribute to the decision to screen that might potentially confound these results?

Errors

- 1) Results, 2nd paragraph, 7th sentence: text lists effect size for *M. curtsii/mulieris* as 0.51 but this is reported as 0.53 in table S6.
- 2) Table S6, rows 14-18 are mislabeled and should be "All visits for AA"
- 3) Methods, 4th paragraph, last sentence: change "1505 samples from 432 subjects" to "1263 samples from 432 subjects"
- 4) Figure 1a, 1b and S1: the number of samples used in each panel aren't consistent. For example in 1a the sum of AA and non-AA for all visits is 1503 not 1505, the sums between 1a and 1b don't match for visit 1 all subjects and the lower left panel of 1b should sum to 1107 not 390. Figure S1a left panel should sum to 503.
- 5) Figure 1c&d (right) & figure S2, what do the n's reported in the top of the plots refer to and why do they change for each taxa?

Point by point response to reviewers' comments:

Reviewer #1 (Remarks to the Author):

The authors present the results of their nested case control study designed to better understand the relationship between the cervicovaginal microbiome and immune response and preterm birth. The particular relevance of this topic to the broader audience should not be underestimated. While the authors correctly identify the short term sequelae of preterm birth they fail to mention the long term health risks associated with preterm birth for both the mother and fetus.

Answer: The introduction now reads (lines 33-43):

“PTB occurs in 1 out of every 10 pregnant women in the United States and over 65-75% of all PTBs are spontaneous with the idiopathic onset of cervical change, uterine contractility and/or rupture of fetal membranes, while the remaining preterm births are medically indicated for reasons such as preeclampsia or fetal distress³. The economic burden of preterm birth is staggering, with an estimated cost of \$26 billion dollars per year in the United States alone^{4,5}. While there is known racial disparity in spontaneous preterm birth (sPTB) with African-American women having significantly higher rates than non-African American women, factors that underpin this disparity remain elusive.⁶ While there are medical, societal and economic costs to the actual PTB, the larger cost to our society stems from the need for long-term care for these preterm infants.⁷ Ex-preterm children are at increased risk for a spectrum of neurobehavioral disorders—ranging from cognitive deficits to cerebral palsy to neurobehavioral abnormalities including autism.⁸⁻¹⁰”

Understanding the etiology, identifying markers of risk and developing therapeutic strategies could have substantial impact on health care delivery in the US.

The manuscript is well written, and the results advance our understanding of potential mechanisms that result in preterm birth as well as opportunities for biomarker discovery and therapeutic interventions. There are several concerns that should be addressed:

1. The use of cervicovaginal community state types (CSTs) has been a common approach for evaluation of the vaginal microbiome but also recently challenged. The approach of categorizing data based on prior published work may limit discovery of new or novel findings.

Answer: We use community state types (CST) as a way to reduce dimensionality of a complex dataset. Using CST is extremely valuable when studying the vaginal microbiota as they do represent meaningful biology unlike in other body sites. The analysis afforded us the identification of difference in the vaginal microbiota structure in African American women, which led us to fully consider race/ethnicity as a major factor in the rest of our analysis. CST analysis allowed us to show that the risk for preterm birth contributed by the vaginal microbiota is different in African American than non-African American. That said, while we do present CST level analysis, we perform a more complex phylotype level analysis and identified specific taxa associated with the increase risk of PTB, which is further shown to be modulated by human beta-defensin 2 level.

2. The authors use the term “cervicovaginal” in vaginal self-collected samples. It is not clear how they can be certain that the cervix is part of the sampling. Cervical microbiome warrants separate evaluation.

Answer: We have to respectfully disagree with this comment. The mucus that is sampled on vaginal wall is mostly made of mucus that originates from the cervix and flows down. Our work has clearly showed that there is no difference in the composition and structure of the microbiota in samples collected from the vaginal wall and the endocervix. Fluids in the vagina are not restricted anatomically to one area or another in the vagina. Thus, when a woman self-collects a vaginal swab, she actually collects cervicovaginal microbiota.

3. Although not clearly stated, it is assumed that race is defined by patient self-report. This definition can be problematic and analysis specifically designed to evaluate for differences in racial groups may introduce bias.

Answer: We agree, that self-report of race/ethnicity is not ideal. Ideally, a genetic screen would assign each participant to a race/ethnicity or admixture, but this would require a major and expensive undertaking. In

addition, in the context of a future diagnostic/predictive algorithm, it would be prohibitive at this stage to perform a genetic screen. Reports have shown that the level of mis-self-assignment is actually very low. Further, our data for the cohort are consistent with the general population seen at UPENN.

That said, we have clarified that race was self-reported in the text which now reads on line 245-246: *“Controls were frequency matched by self-reported race to the cases.”*

4. Other variables that might impact the microbiome are not considered (vaginal douching, oral sex etc)

Answer: While this is interesting, and other studies (including ours) have addressed these factors, if these behaviors affect the microbiota it is captured by our case-control study design. Table S1 describes some of these factors, and it is important to note that only about 1% of the participants used vaginal douches, which excludes this factor as associated with our outcome. Sexual behaviors were recorded and were not shown to be different between the cases and controls.

5. The authors present the results of spontaneous preterm birth but do not consider whether those spontaneous preterm births were a result of spontaneous preterm labor or preterm premature rupture of membranes.

Answer: Only women presenting with cervical change and/or PPRM were classified as spontaneous preterm birth. While it has been suggested that there is a clinical distinction in these two entities, clinically it is difficult to precisely characterize patients. For example, if a woman presents at 3 cm dilated with PPRM, is she considered “preterm labor” or PPRM. For our study, we recorded a patient with this scenario as both. Since the clinical distinction/phenotype is not clear, we did not feel it was prudent to perform a sub-analyses on this clinical metric.

Reviewer #2 (Remarks to the Author):

Spontaneous preterm birth (sPTB) proportionally affects African American (AA) women. Elovitz et al. describe a very large study that tries to unravel (sic!) the many factors that might play a role in this tragic statistic. In this study of nearly 2000 women (three quarters of which were AA), microbial (vaginal microbiome composition and absolute bacterial load) as well as immunological (beta-defensin-2) were investigated. The main conclusions are:

- * Lower levels of Lactobacillus spp. is associated with sPTB in non-AA women but not in AA women.
- * Several taxa, but in particular a high abundance of Mobiluncus curtisii/mulieris, were associated with an increased risk for sPTB
- * A decrease of bacterial absolute abundance after 24 weeks of gestation was found in women who delivered preterm.
- * High lactobacillus levels reduced the probability of sPTB in the presence of high Mobiluncus curtisii/mulieris
- * AA women who had sPTB had lower levels of b-defensin-2. However, for nonAA women, this was the other way around.
- * in AA women the risk of sPTB is high even in the presence of Lactobacillus spp. when b-defensin-2 levels are low

These results illustrate the complex interaction between potential pathogens, beneficial Lactobacillus, immune factors, and ethnicity. Even within this very large cohort, patterns are not completely clear. However, this study is an enormous step forward in the search for factors that can be used to predict whether or not a pregnancy will be term or not.

The text is very well written, and the figures are mostly clear, albeit dense and sometimes hard to read. Some suggestions for textual and visual improvements are written below.

Introduction

1. Line 32: Every year: Please add if this number is worldwide or in the US

Answer: The sentence on line 32 now reads:

“Every year worldwide, 1.1 million babies die from consequences of prematurity.”

2. L33. It might be helpful to add what the other 25% of PTB are, since readers (including me) might assume that nearly all PTB are spontaneous.

Line 33-36 now reads:

“PTB occurs in 1 out of every 10 pregnant women in the United States and over 65-75% of all PTBs are spontaneous with the idiopathic onset of cervical change, uterine contractility and/or rupture of fetal membranes, while the remaining preterm births are medically indicated for reasons such as preeclampsia or fetal distress³.”

Results

3. L84. It came as a surprise that only 14 positive controls were used to find taxa associated with PTB when 107 sPTB controls were available. Maybe I am not understanding this model correctly - is this similar to machine learning where a subset is used for training and the remaining samples for confirmation? The description in the Methods was way over my head but some reasoning for only using this small subset would be welcome. When all women were considered, the model found fewer taxa. How was this subset chosen, randomly?

Answer: As stated on lines 98-99, the 14 controls were sequencing positive controls and not controls included in our case-control study design, in which 432 control women were included matching the 107 cases. The 14 controls are vaginal samples with known composition samples that are sequenced on each plate of 90 samples and are used as positive controls to evaluate potential technical errors. In this study, we introduce a major novel use of these positive controls. We combine these controls from the 14 plates of samples we sequence and model error measurements (i.e., the probability to detect a low abundance taxon each time a sample is sequenced). This error measurement model was incorporated into our Bayesian model of the dependence of the risk of sPTB on the relative abundance of specific bacterial taxa. See also point 7 below.

To clarify the fact that these controls are “sequencing positive controls” line 91 now reads:

“and modeled using a dataset generated from 14 sequencing positive control samples (Supplementary Table 5).”

In addition, the method section was clarified and now reads on lines 267-270: “Additionally, a positive control composed of a mixture of 20 vaginal biological specimens of known composition combined into one tube was processed and sequenced in parallel on each of 14 pools of study samples as per the laboratory standard protocol (Supplementary Table 5).”

4. L90 "for each taxa" - should be "for each taxon"

Answer: Corrected, now on line 97.

Methods.

5. L220. The wording is not clear here. I assume that samples were not "self collected by the research coordinator" - so rewording would help.

Answer: We apologize for this error; the samples were indeed self-collected by the participants during their clinical visit or collected by a research coordinator if a clinical exam was indicated.

The text now on lines 228-230 now reads:

“Cervicovaginal specimens were self-collected by the participant or collected by a research coordinator if a clinical exam was indicated at 3 different prenatal visits:...”

6. L238. "MoBio Microbiome kit" - please provide manufacturer. Is there a more specific name for this kit?

Answer: The text now on lines 249-251 now reads:

“ESwabs were thawed on ice, and 300 μ l of Amies transport medium containing vaginal secretion were processed using the MoBio PowerMag Microbiome DNA/RNA kit (now MagAttract PowerMicrobiome DNA/RNA kit, Qiagen) automated on a Hamilton Microlab STAR robotic platform...”

7. L254. "a set of 14 positive controls composed of a mixture of vaginal biological specimen combined into one tube" was not very clear. Were 14 samples pooled into one tube? Or were there 14 samples that each consisted of a mixture of specimens? How many specimens? Should "specimen" be "specimens" here? Please reword to make it more clear what was done.

Answer: The text was clarified and now reads on lines 267-270:

“Additionally, a positive control composed of a mixture of 20 vaginal biological specimens of known composition combined into one tube was processed and sequenced in parallel on each of 14 pools of study samples as per the laboratory standard protocol (Supplementary Table 5).”

8. L296. "Swabs were washed in sterile PBS" is not clear. Do the authors mean that the swab was soaked (not washed) in PBS, and the PBS then was analyzed? Or did they wash the swabs with PBS, then discard the PBS and analyze the swabs?

Answer: The text now reads on line 309: “Materials on the swabs were eluted in sterile PBS with a protease inhibitor cocktail (Complete Mini) for 5 minutes to release the soluble proteins. The eluate solutions were then analyzed for the presence...”

9. L306. "All scripts used in statistical analyses are available in the Supplemental data" - I could not find these in the materials shared for peer review.

Answer: The statement was removed as it was redundant with the section Data and materials availability where it says:

“All analysis scripts used in the analysis used in this manuscript are available at: https://github.com/ravel-lab/M_and_M”

10. L323. The description of the "Bayesian logistic regression nonparametric adaptive smoothing model" was beyond my expertise and I could not peer review this section. The method described here appears to be a modification of an R package but I would not have any idea how to perform this.

*Answer: The non-parametric adaptive spline part of the model was adapted from the *spmrf* R package cited in reference 38. The model was built for this specific analysis. The code is available on the GitHub site listed in the section Data and materials availability.*

11. L375. Sequence data was not available for peer review.

Answer: The sequence data has been submitted to SRA and dbGaP along with the metadata associated with the study. The accession number is still pending. The dbGaP deposition is taking a lot longer than expected unfortunately, but is a requirement of the funding agency (NIH).

Figures and tables

12. Figure 1. Some of the text is very small. The quality of the graph is excellent, so I can zoom in and read it, but it might be worth investigating if some text could be printed a bit larger.

Answer: Some of the text was increased in size in Figure 1a,b, We will discuss the appropriateness of the font size with the editor and modify the figure accordingly

13. Figure 1e. I assume RA stands for "relative abundance", but it would be helpful if that could be defined in the legend text.

Answer: RA is now defined in the figure legend, which now reads on line 539-542:

*“(e) Kaplan-Meier survival plot for BVAB3, *M. curtisii*/mulieris and *S. sanguinegens* in all and AA women who harbor these bacterial taxa at relative abundance (RA) below (blue) or above (orange) the threshold values above which the risk of sPTB is significant different from baseline.”*

14. Figure 2a. It took a while to correctly interpret the 3 Lactobacillus graphs, partially because the "tertile" term might not be familiar for all. Would it be more clear if the titles were "Low Lactobacillus", "Medium Lactobacillus", "High Lactobacillus", with additional explanation about the tertiles in the text? The authors can ignore this if this is not scientifically correct.

Answer: We have clarified the headers on figure 2a, which now reads: Low Lactobacillus (tertile 1), Medium Lactobacillus (tertile 2) and High Lactobacillus (tertile 3).

15. Figure 2a. Since the X axes are the same for all 4 panels, but not clearly labeled, it might be more clear to call them "log10 (Relative abundance of *Mobiluncus curtisii*/mulieris) - the title of the axis might only be shown once. Just a suggestion to make it more clear that the axes shown are the same for all 4 graphs.

Answer: The axis label was changed. Some of the text was also increased in size for improved readability.

16. Figure 2b. Assuming that the first 2 datasets shown here (in black) are the combined samples irrespective of CST, it might be more clear to say "all" or "combined" under the graph.

Answer: the label "All CSTs" was added to figure 2b.

17. Figure 3. I compliment the authors for this figure, which conveys a lot of information in single graph, and was very interesting to explore. However, I was not sure how to interpret the legend shown at the top. Here the authors want to add the abundance of *Mobiluncus* (high or low) but it is very unclear. Is only the size (small or big) and shape (triangle/square) important or also the outline and shading color? Should I be looking for large and small squares and triangles, or for grey or black shapes? The smaller "square" on the right appears to be shaded with an outline and an X in the middle, and is also not really a square but a rectangle. It was also not clear why all triangles had an outline and none of the squares. What do the high-abundance *Mobiluncus* triangles and squares have in common; their shading? This might need some revision.

Answer: The issue was fixed. The shapes in the legend above the figure and for the CST shouldn't have a X but be solid color. The outline on the triangle are just to highlight them as they represent sPTB. The legend was modified to indicate that RA means relative abundance (line 560).

18. Figure S3. I was not sure how to interpret this graph. Is the X axis showing the timepoint during pregnancy at which the sample was taken? Or is the X axis showing the number of weeks at which the baby was born (probably not since both Term and sPTB are shown?) I suspect it is the first, but it would help to better word what is shown here.

*Answer: The X-axis indicates the gestational age for the corresponding sample that fits within the curve. We collected samples within three windows (16-20, 20-24 and 24-28 weeks of gestation) and these represent the range of gestational ages shown on the graph (x-axis is 16-28 weeks).
On Figure S3, the font of some of the text was increased for readability.*

Reviewer #3 (Remarks to the Author):

This paper evaluates the association between vaginal microbiome and an immune marker with spontaneous pre-term birth. This is a well written paper using data from a large, prospective case-control cohort. Major strengths include the longitudinal collection of samples, inclusion of negative and positive controls, estimation of total bacterial burden and the approach taken to classify taxa. The authors have included a large amount of

relevant supplementary material and publicly posted their code and data to better enable reproducibility of results. I offer some suggestions for the improvement of the paper.

Major

1) A strength of the study is the longitudinal design. The logistic regression includes a random effect which is presumably a random subject effect to account for repeated measures. This appears to be the only analysis in which the repeated measures were accounted for in the model. Accounting for repeated measures in all of the analyses using data from all visits might be especially important since controls were more likely to have all three samples collected whereas the sPTB groups had a more even distribution of 1,2 or 3 samples collected. Although this concern is somewhat tempered by the inclusion of analyses for each visit separately. Regardless, the ability to evaluate changes over time within a woman are not capitalized in the current analyses and there is the potential for missing important trends over time, for example there are women who switch CSTs over time, might this reflect important changes in addition to simply noting their current CST?

Answer: The dependence of samples collected within a woman over time is accounted for in the models. Mixed effect models (accounting for repeated measures) were used only in the context of CST analyses as in the case of phylotype level analyses investigating differences in relative abundances of a given taxa within sPTB vs TERM subjects (calling for mixed effect models) only applies in a situation where distribution of relative abundances is unimodal, which is not the case for many taxa in human vagina. This is why we used adaptive spline models estimating probability of sPTB as a function of log relative abundance of a given phylotype. These models address directly the central question of risk of sPTB and also do not require use of mixed effect terms as the birth status is constant within each subject. As for the evaluation of changes over time, we have performed analyses of changes in frequencies of CST between different visits. We have not detected any significant signal as the study was not powered enough to answer this kind of question (only 5% of the women in the nested case-control study switch CSTs at least once).

2) The effect of missing data is not addressed, while it may be assumed that some of the sample are missing at random due to inadequate samples or simply missed visits, it is also likely that some of visit 3 samples in the sPTB group are missing not at random depending on when the sPTB event occurred. Given this is already a complex series of analyses, I think it would be helpful to at least acknowledge this issue and briefly discuss potential implications.

Answer: Clearly, frequency of sPTB events goes down with gestation age, this is why we have emphasized analyses of samples collected at V1 as they are the most important not only from the clinical stand point (early detection of preterm birth), but also due to deflation of sPTB frequencies at V2 and V3 visits. Our analysis addresses the issue of the dependence between frequency of sPTB and gestation age, by modeling the risk of sPTB using adaptive spline models, where the dependence of sPTB frequencies on gestation age is not an issue.

3) In the description of the survival analysis it is not clear if data from all visits were included and if so, were the taxa included as time varying covariates? Was there any weighting or a frailty term included to account for the matched case-control design?

Answer; In the survival analyses data from all visits were used, however, participants were included in one of the two groups if their vaginal microbiota harbor the taxa of interest above or below the threshold established on figure 1. Taxa were included as time varying covariates. The reported p-values are from models without frailty term as for 4 out of 6 taxa for which the survival modeling was done the models with frailty term failed to converge.

Minor

1) What was the definition of sPTB?

Answer: The definition of sPTB was “women who presented with either cervical dilation and/or premature rupture of membranes and delivered prior to 37 weeks. A woman being delivered for maternal indications (e.g.

preeclampsia) or fetal indication (e.g. fetal growth restriction) were not considered spontaneous preterm births.”

We have added some text on Line 234-239 and now reads:

“Subjects were followed to delivery. All delivery outcomes were recorded. Cases of preterm birth were adjudicated by the PI to determine if the cases were medically indicated preterm births or spontaneous. PTB was considered spontaneous when a woman presented with either cervical dilation and/or premature rupture of membranes and delivered prior to 37 weeks. Women delivered for maternal indications (e.g. preeclampsia) or fetal indication (e.g. fetal growth restriction) were not considered spontaneous preterm births.”

2) Table S1 shows that BMI is stable across the 3 visits, is this baseline BMI or is it adjusted for expected weight gain during pregnancy?

Answer: The BMI shown on Table S1 is the actual BMI measured during a clinical visit during the time period of interest, no adjustments were performed. Standard protocols were used to measure BMI. It was measured based on height (at the start of pregnancy) and weight during that sample period. So individual women might have gained or lost weight, but the mean BMI did not change by case-control status. The sampling period was over 12 weeks and considering that most weight gain usually occurs after 28 weeks, one would not have expected to observe major change in BMI between 16 and 28 weeks of gestation.

3) Results 5th paragraph: what is the distribution of RA of *M. curtisii/mulieris* in each of the *Lactobacillus* tertiles? Given that the relative abundances are used, an increase in *Lactobacillus* must necessarily result in a decrease of something else, might this result simply be an artefact of compositional data?

*Answer: The ranges of \log_{10} relative abundances of *M. curtisii/mulieris* in each of the *Lactobacillus* tertiles are between -4 and -1. Kolmogorov-Smirnov tests comparing \log_{10} relative abundances of *M. curtisii/mulieris* in each of the *Lactobacillus* tertiles are all above 0.8 (the comparisons were between all pairs of *Lactobacillus* tertiles). Thus, there is no significant differences between distributions of *M. curtisii/mulieris* relative abundance in these tertiles. However, other taxa certainly changed based on the compositional nature of the data analyzed.*

4) The decrease in the total bacterial burden is an interesting finding. Is this decrease observed over time within women or could this be an artefact of missing visit 3 samples in women who had a PTB?

Answer: The following figure shows a scatter plot of \log_{10} bacterial absolute abundance values vs gestation age (GA). As we can see there is a clear pull towards low bacterial absolute abundances among sPTB subjects at GA weeks 25,26,27. The table below the figure shows p-values of t-test comparing \log_{10} bacterial absolute abundances between TERM and sPTB subjects within a specified GA week. Non-significance of the test at week 27 is most likely due to smaller number of sPTB samples than in previous weeks. Thus, the trend of lower bacterial absolute abundances among sPTB subjects is driven by a subpopulation of sPTB except at the 27 weeks of GA in this dataset.

[16,] 0.95582549
 [17,] 0.42392852
 [18,] 0.57114902
 [19,] 0.14636827
 [20,] 0.74776194
 [21,] 0.63574469
 [22,] 0.58753468
 [23,] 0.27869531
 [24,] 0.77982558
 [25,] 0.03798337
 [26,] 0.02500201
 [27,] 0.07227718

5) Is a difference of 0.3 logs for beta-defensin-2 clinically meaningful? With a large sample size, small effect sizes can be statistically significant but may not be clinically different. This is also true of the individual taxa selected, the median relative abundance is very small.

Answer: It is difficult to interpret the clinical significance of any difference until these biomarkers are validated in another clinical trial outside of the case-control study. Further, because this is the first report of microbial-immune correlation, it is difficult to contextualize and interpret without a validation. In addition, Figure S6 suggests the difference might be important as these differences are higher between races.

6) What is the reasoning behind removing samples with fewer than 1000 sequences, was the total bacterial burden low for these samples?

Answer: This is a common practice in the field of microbiome analyses. These samples usually represent suboptimal processing. Low sequence count is in general a reflection of sequencing process random variation (including DNA quality, etc) and setting a threshold for sample read count reduces relative abundance inflation bias for highly abundant taxa due to non-detection of low abundance taxa.

7) The methods indicate that SLPI was also measured but these results aren't presented.

Answer: This was an error, and SLPI was not measured. The sentence referring to SLPI was removed.

8) The statistical analysis mentions that the logistic regression for sPTB controls for race and birth outcome. What is birth outcome and how does this differ from sPTB?

Answer: In the manuscript, we mention controlling for race and birth outcome on lines 319-323. in the context of CST frequencies analysis, not sPTB, using logistic regression. “Mixed effects logistic regression models were used to assess difference in the frequency of CSTs in women who delivered at term and those who delivered preterm, while controlling for race or birth outcome (term vs sPTB).”

The birth outcome is indeed sPTB.

9) How were the 0's handled prior to the log transformation for the relative abundance?

Answer: In all analyses of dependence on the frequency of sPTB as a function of log relative abundance of a given phylotype, 0's were removed from the analysis as it does not make sense to include them where we ask how the frequency of sPTB depends on relative abundance of a given phylotype. We have performed analyses of dependence of sPTB frequencies on presence/absence of a given phylotype but have not seen any significant signals.

10) Figure 3 & S7, it is difficult to distinguish points above and below the *M. curtsii/mulieris* cutoff.

*Answer: We have adjusted the size of the larger triangle and square which indicates *M. curtsii/mulieris* above the threshold.*

11) Table 1: It seems odd to me that the controls would be screened more often for cervical length compared to women with sPTB, is this because these women were more likely to have a history of previous sPTB? Are there other factors that contribute to the decision to screen that might potentially confound these results?

Answer: An Asterix has been included in Table 1 next to the label “Cervical length screening performed at Level II ultrasound” that reads (Excel file and Line 567): “ *women with prior PTB are screened prior to level II ultrasound (16-22 weeks) and not all women do undergo screening again during the specific level II ultrasound (19-21 weeks) which is performed to assess fetal anatomical structures.”*

Errors

1) Results, 2nd paragraph, 7th sentence: text lists effect size for *M. curtsii/mulieris* as 0.51 but this is reported as 0.53 in table S6.

Answer: Corrected in the text. It should have been 0.53.

2) Table S6, rows 14-18 are mislabeled and should be “All visits for AA”

Answer: Corrected in Supplementary Table 6.

3) Methods, 4th paragraph, last sentence: change “1505 samples from 432 subjects” to “1263 samples from 432 subjects”

Answer: Corrected in the text. It should have been “1263 samples from 432 subjects.”

4) Figure 1a, 1b and S1: the number of samples used in each panel aren't consistent. For example in 1a the sum of AA and non-AA for all visits is 1503 not 1505, the sums between 1a and 1b don't match for visit 1 all subjects and the lower left panel of 1b should sum to 1107 not 390. Figure S1a left panel should sum to 503.

Answer: Corrections were made to the figure to reflect the correct number of samples included in the analyses shown in each panel of Figure 1a and 1b, as well as Supplemental Figure S1a and S1b.

5) Figure 1c&d (right) & figure S2, what do the n's reported in the top of the plots refer to and why do they change for each taxa?

Answer: The n in the top of each plot represent the number of samples containing the phylotype analyzed and included in the analysis. The figure legends have been corrected and now reads:

Reviewers' comments:

Reviewer #1 (Remarks to the Author):

The reviewers have adequately responded to concerns raised by this reviewer with one exception. It was previously noted that the authors present the results of spontaneous preterm birth but do not consider whether those spontaneous preterm births were a result of spontaneous preterm labor or preterm premature rupture of membranes. It is true that there is likely overlap in these clinical phenotypes. It has been suggested that preterm premature rupture of membranes (PPROM) is linked more closely with infection and not leveraging this dataset to evaluate this question is a missed opportunity. The patient that presents with preterm contractions and delivers is not the same clinical phenotype as a patient with PPRM that remains pregnant for weeks after rupture of membranes. It is possible that the numbers of preterm birth will not be sufficient to allow for this analysis. The authors have the expertise to attempt to categorize these deliveries and should do so.

Reviewer #2 (Remarks to the Author):

The authors have addressed all the issues raised by the other 2 reviewers and by me. One very minor issue on the revised version (which should be easy to address):

L37. "with an estimated cost of \$26 billion dollars per year in the United States alone"

There is both a dollar sign and the word "dollars" - one of these can probably be deleted.

I appreciate the effort and additional analyses that the authors performed to address some of my previous comments. I do have some remaining suggestions and apologize if my first round of comments were not clearly specified. I think this is an important paper and want to see it published in this journal. Given it is likely to have high-impact and serve as an example of future analyses of similar studies I think it is important to have the methods clearly described. I have copied my original comments (black) and the author's responses (blue italic) along with my additional suggestions (red) for context for those points where I had additional suggestions, other previous comments were omitted for brevity.

Reviewer #3 (Remarks to the Author):

This paper evaluates the association between vaginal microbiome and an immune marker with spontaneous pre-term birth. This is a well written paper using data from a large, prospective case-control cohort. Major strengths include the longitudinal collection of samples, inclusion of negative and positive controls, estimation of total bacterial burden and the approach taken to classify taxa. The authors have included a large amount of relevant supplementary material and publicly posted their code and data to better enable reproducibility of results. I offer some suggestions for the improvement of the paper.

Major

- 1) A strength of the study is the longitudinal design. The logistic regression includes a random effect which is presumably a random subject effect to account for repeated measures. This appears to be the only analysis in which the repeated measures were accounted for in the model. Accounting for repeated measures in all of the analyses using data from all visits might be especially important since controls were more likely to have all three samples collected whereas the sPTB groups had a more even distribution of 1,2 or 3 samples collected. Although this concern is somewhat tempered by the inclusion of analyses for each visit separately. Regardless, the ability to evaluate changes over time within a woman are not capitalized in the current analyses and there is the potential for missing important trends over time, for example there are women who switch CSTs over time, might this reflect important changes in addition to simply noting their current CST?

Answer: The dependence of samples collected within a woman over time is accounted for in the models. Mixed effect models (accounting for repeated measures) were used only in the context of CST analyses as in the case of phylotype level analyses investigating differences in relative abundances of a given taxa within sPTB vs TERM subjects (calling for mixed effect models) only applies in a situation where distribution of relative abundances is unimodal, which is not the case for many taxa in human vagina. This is why we used adaptive spline models estimating probability of sPTB as a function of log relative abundance of a given phylotype. These models address directly the central question of risk of sPTB and also do not require use of mixed effect terms as the birth status is constant within each subject. As for the evaluation of changes over time, we have performed analyses of changes in frequencies of CST between different visits. We have not detected any significant signal as the study was not powered enough to answer this kind of question (only 5% of the women in the nested case-control study switch CSTs at least once).

Response: I'm not sure I entirely understand the authors response, as the correlation between repeated samples collected within a subject were not accounted for in all models, only the mixed effect logistic regression models. The use of mixed models could be employed to account for repeated measures regardless of the distribution of the relative abundance as generalized linear mixed models and mixed effects spline models are standard, albeit complex, analytic approaches and have been previously applied to microbiome data. My concern was that any correlation between observations collected on the same subject be accounted for in the model. I

recognize that the authors already have complex analyses (splines, non-normal outcome distributions) included and that extending all analyses to incorporate a random subject effect or to use generalized estimating equations would be a major revision. I think simply and clearly stating which analyses assume data from all visits within a subject are independent would be sufficient to allow the reader to put the results in context.

2) The effect of missing data is not addressed, while it may be assumed that some of the samples are missing at random due to inadequate samples or simply missed visits, it is also likely that some of visit 3 samples in the sPTB group are missing not at random depending on when the sPTB event occurred. Given this is already a complex series of analyses, I think it would be helpful to at least acknowledge this issue and briefly discuss potential implications.

Answer: Clearly, frequency of sPTB events goes down with gestation age, this is why we have emphasized analyses of samples collected at V1 as they are the most important not only from the clinical stand point (early detection of preterm birth), but also due to deflation of sPTB frequencies at V2 and V3 visits. Our analysis addresses the issue of the dependence between frequency of sPTB and gestation age, by modeling the risk of sPTB using adaptive spline models, where the dependence of sPTB frequencies on gestation age is not an issue.

Response: I agree that placing emphasis on the V1 samples is appropriate. An adaptive spline accounts for non-linear dependence between the risk of sPTB and gestational age but it still assumes that data are missing at random. It uses the observed data at the higher gestational ages to make inferences about the missing values. Given that the data for V3 are likely not missing at random but missing because the subject experienced a PTB then this assumption is invalid. Although there are missing data methods that can evaluate these assumptions, again, I think simply acknowledging this may be an issue in the manuscript would be sufficient.

3) In the description of the survival analysis it is not clear if data from all visits were included and if so, were the taxa included as time varying covariates? Was there any weighting or a frailty term included to account for the matched case-control design?

Answer; In the survival analyses data from all visits were used, however, participants were included in one of the two groups if their vaginal microbiota harbor the taxa of interest above or below the threshold established on figure 1. Taxa were included as time varying covariates. The reported p-values are from models without frailty term as for 4 out of 6 taxa for which the survival modeling was done the models with frailty term failed to converge.

Response: Please include description of the inclusion of taxa as time varying covariates in the methods.

Minor

4) The decrease in the total bacterial burden is an interesting finding. Is this decrease observed over time within women or could this be an artefact of missing visit 3 samples in women who had a PTB?

Answer: The following figure shows a scatter plot of log₁₀ bacterial absolute abundance values vs gestation age (GA). As we can see there is a clear pull towards low bacterial absolute abundances among sPTB subjects at GA weeks 25,26,27. The table below the figure shows p-values of t-test comparing log₁₀ bacterial absolute abundances between TERM and sPTB subjects within a specified GA week. Non-significance of the test at week 27 is most likely due to smaller number of sPTB samples than in previous weeks. Thus, the trend of lower bacterial absolute abundances among sPTB subjects is driven by a subpopulation of sPTB except at the 27 weeks of GA in this dataset.

[16,] 0.95582549

[17,] 0.42392852
[18,] 0.57114902
[19,] 0.14636827
[20,] 0.74776194
[21,] 0.63574469
[22,] 0.58753468
[23,] 0.27869531
[24,] 0.77982558
[25,] 0.03798337
[26,] 0.02500201
[27,] 0.07227718

Response: I apologize if this original comment was unclear. I meant to suggest that some additional analysis would help determine whether the difference is due to a decrease in load (as suggested by the heading in the results) or whether those women with higher gestational ages in the sPTB group with low load are driving the difference. This can be determined by 1) re-doing the analysis above but only including those women with later GA samples and 2) looking at subject-specific trends over time within women in each group. If the subject specific plots indicate that the trajectories over time are relatively flat within each women then difference observed at the later times is not due to an actual decrease over time. Alternatively, the authors could simply change to heading of this section to “Bacterial taxa AA is lower after 24 weeks...” to better reflect that the difference may not be due to a decrease over time.

5) Is a difference of 0.3 logs for beta-defensin-2 clinically meaningful? With a large sample size, small effect sizes can be statistically significant but may not be clinically different. This is also true of the individual taxa selected, the median relative abundance is very small.

Answer: It is difficult to interpret the clinical significance of any difference until these biomarkers are validated in another clinical trial outside of the case-control study. Further, because this is the first report of microbial immune correlation, it is difficult to contextualize and interpret without a validation. In addition, Figure S6 suggests the difference might be important as these differences are higher between races.

Response: If there is room, it would be helpful if points made in the author's response could be added to the text.

6) What is the reasoning behind removing samples with fewer than 1000 sequences, was the total bacterial burden low for these samples?

Answer: This is a common practice in the field of microbiome analyses. These samples usually represent suboptimal processing. Low sequence count is in general a reflection of sequencing process random variation (including DNA quality, etc) and setting a threshold for sample read count reduces relative abundance inflation bias for highly abundant taxa due to non-detection of low abundance taxa.

Response: In my experience, these samples are only removed if the total bacterial load is low, if there is sufficient load but low number of sequences the sample is re-sequenced. I wasn't sure if this was the case in this study. I have no further suggestions on this point.

9) How were the 0's handled prior to the log transformation for the relative abundance?

Answer: In all analyses of dependence on the frequency of sPTB as a function of log relative abundance of a given phylotype, 0's were removed from the analysis as it does not make sense to include them where we ask how the frequency of sPTB depends on relative abundance of a given phylotype. We have performed analyses of dependence of sPTB frequencies on presence/absence of a given phylotype but have not seen any significant signals.

Response: Although I think one could make a compelling argument why those samples with 0's do carry information and shouldn't be excluded, at the very least the exclusion of these samples should be clarified in the text so that the reader understands what was done and can evaluate results appropriately within this context.

Errors

5) Figure 1c&d (right) & figure S2, what do the n's reported in the top of the plots refer to and why do they change for each taxa?

Answer: The n in the top of each plot represent the number of samples containing the phylotype analyzed and included in the analysis. The figure legends have been corrected and now reads:

Response: Please revise the figure legends to explain the n displayed in the top of each plot.

Additional Suggestion:

- 1) "All scripts used in statistical analyses are available in the Supplemental data" – remains in statistical analysis section despite response to reviewer 1, please remove as reader is referred to the Github repository in the Methods. I very much appreciate the posting of the code and think this will be a useful reference for readers.
- 2) From the code, it looks as though the mixed effect logistic regression was actually approximated with a Poisson distribution. Please clearly state this in the methods.

January 24, 2019

Response to reviewers: Response to new comments are in **red**, previous comments are in black, and previous answers in **blue**.

Reviewer #1: *The authors have adequately responded to concerns raised by this reviewer with one exception. It was previously noted that the authors present the results of spontaneous preterm birth but do not consider whether those spontaneous preterm births were a result of spontaneous preterm labor or preterm premature rupture of membranes. It is true that there is likely overlap in these clinical phenotypes. It has been suggested that preterm premature rupture of membranes (PPROM) is linked more closely with infection and not leveraging this dataset to evaluate this question is a missed opportunity. The patient that presents with preterm contractions and delivers is not the same clinical phenotype as a patient with PPROM that remains pregnant for weeks after rupture of membranes. It is possible that the numbers of preterm birth will not be sufficient to allow for this analysis. The authors have the expertise to attempt to categorize these deliveries and should do so.*

Answer from authors:

The proposed association between PPROM and overt infection of the uterine/amniotic cavity is confounded by the fact that clinically, some women with PPROM have a latency period before delivery occurs. Some women will deliver in 24 hours after PPROM. Most women with PPROM will deliver within 7 days. There is a small group of women who present with PPROM who have a prolonged latency period and may not deliver for a few weeks. The reports of amniotic infection, as documented by culture or PCR, in these women, may be causative in the PPROM or, it may represent a secondary infection from delay in delivery with unsealed membranes.

As reviewer #1 noted, our team is well positioned to adjudicate clinical phenotypes of preterm birth. However, the phenotypic distinction between PPROM and spontaneous labor can be quite difficult. If a patient presents with contractions and is 3 cm dilated with PPROM, is that patient classified as PPROM or spontaneous labor or both? The authors believe that the phenotypic distinction is not as clear as proposed. For those few women who have a latency period of weeks before delivery, a phenotypic differentiation is possible, but those patients are few in number. Based on these difficulties and uncertainties surrounding a potential differential diagnostic, we do not believe additional analyses investigating the few cases of PPROM that do not deliver for weeks is warranted and could possibly lead to bias in the conclusions.

That said, we have followed the recommendation of the editor and have included additional text in the discussion on this point in the main manuscript, see below.

As noted on page 8 of the revised manuscript (line 243-246), a clear definition of sPTB including preterm labor and PPROM is provided. "Cases of preterm birth were adjudicated by the PI to determine if the cases were medically indicated preterm births or spontaneous. PTB was considered spontaneous when a woman presented with either cervical dilation and/or premature rupture of membranes and delivered prior to 37 weeks."

In this newly revised manuscript, additional text was added to the discussion stating that preterm labor and PPROM are considered under the umbrella of "sPTB" and thus, treated as the same clinical entity for interventions and prevention strategies.

The discussion now reads (line 210-213):

"Women with a sPTB present with contractions, cervical dilatation and/or preterm premature rupture of membranes. As women can present with several of these symptoms, clear phenotyping by preterm rupture of membranes or cervical dilation is problematic. Therefore, for clinical studies, these various clinical presentations are collectively characterized as sPTB."

Reviewer #3

Comment 1.

Original comment 1: A strength of the study is the longitudinal design. The logistic regression includes a random effect which is presumably a random subject effect to account for repeated measures. This appears to be the only analysis in which the repeated measures were accounted for in the model. Accounting for repeated measures in all of the analyses using data from all visits might be especially important since controls were more likely to have all three samples collected whereas the sPTB groups had a more even distribution of 1,2 or 3 samples collected. Although this concern is somewhat tempered by the inclusion of analyses for each visit separately. Regardless, the ability to evaluate changes over time within a woman are not capitalized in the current analyses and there is the potential for missing important trends over time, for example there are women who switch CSTs over time, might this reflect important changes in addition to simply noting their current CST?

Answer from authors: The dependence of samples collected within a woman over time is accounted for in the models. Mixed effect models (accounting for repeated measures) were used only in the context of CST analyses as in the case of phylotype level analyses investigating differences in relative abundances of a given taxa within sPTB vs TERM subjects (calling for mixed effect models) only applies in a situation where distribution of relative abundances is unimodal, which is not the case for many taxa in human vagina. This is why we used adaptive spline models estimating probability of sPTB as a function of log relative abundance of a given phylotype. These models address directly the central question of risk of sPTB and also do not require use of mixed effect terms as the birth status is constant within each subject. As for the evaluation of changes over time, we have performed analyses of changes in frequencies of CST between different visits. We have not detected any significant signal as the study was not powered enough to answer this kind of question (only 5% of the women in the nested case-control study switch CSTs at least once).

New comment 1: I'm not sure I entirely understand the authors response, as the correlation between repeated samples collected within a subject were not accounted for in all models, only the mixed effect logistic regression models. The use of mixed models could be employed to account for repeated measures regardless of the distribution of the relative abundance as generalized linear mixed models and mixed effects spline models are standard, albeit complex, analytic approaches and have been previously applied to microbiome data. My concern was that any correlation between observations collected on the same subject be accounted for in the model. I recognize that the authors already have complex analyses (splines, non-normal outcome distributions) included and that extending all analyses to incorporate a random subject effect or to use generalized estimating equations would be a major revision. I think simply and clearly stating which analyses assume data from all visits within a subject are independent would be sufficient to allow the reader to put the results in context.

New response from the authors: We have modified the methods to include a statement addressing this issue according to the reviewer's comments (line 353-355). The text now reads (new text is in red):

"Because this dependence a priori can have a complicated shape, a Bayesian logistic regression nonparametric adaptive spline model was used (adapted from the *spmrf* R package⁴³) as the adaptive splines implemented in the *spmrf* package have superior behavior at the extremities of the independent variable value. **In this model, all samples were assumed to be independent.**"

Comment 2.

Original comment 2. The effect of missing data is not addressed, while it may be assumed that some of the samples are missing at random due to inadequate samples or simply missed visits, it is also likely that some of visit 3 samples in the sPTB group are missing not at random depending on when the sPTB event occurred. Given this is already a complex series of analyses, I think it would be helpful to at least acknowledge this issue and briefly discuss potential implications.

Answer from authors: Clearly, frequency of sPTB events goes down with gestation age, this is why we have emphasized analyses of samples collected at V1 as they are the most important not only from the clinical stand point (early detection of preterm birth), but also due to deflation of sPTB frequencies at V2 and V3 visits. Our analysis addresses the issue of the dependence between frequency of sPTB and gestation age, by modeling the risk of sPTB using adaptive spline models, where the dependence of sPTB frequencies on gestation age is not an issue.

New Comment 2: I agree that placing emphasis on the V1 samples is appropriate. An adaptive spline accounts for non-linear dependence between the risk of sPTB and gestational age but it still assumes that data are missing at random. It uses the observed data at the higher gestational ages to make inferences about the missing values. Given that the data for V3 are likely not missing at random but missing because the subject experienced a PTB then this assumption is invalid. Although there are missing data methods that can evaluate these assumptions, again, I think simply acknowledging this may be an issue in the manuscript would be sufficient.

New answer from authors: We have added text at the end of the statistical method section that reads (line: 407-409)

“Note: In the above analyses, when samples from all three visits are included, given the number the samples is decreasing at each visit due to preterm delivery events, we faced the issue of missing data. The models do not account for the non-randomness of the missing data.”

Comment 3.

Original comment 3. In the description of the survival analysis it is not clear if data from all visits were included and if so, were the taxa included as time varying covariates? Was there any weighting or a frailty term included to account for the matched case-control design?

Answer from authors: In the survival analyses data from all visits were used, however, participants were included in one of the two groups if their vaginal microbiota harbor the taxa of interest above or below the threshold established on figure 1. Taxa were included as time varying covariates. The reported p-values are from models without frailty term as for 4 out of 6 taxa for which the survival modeling was done the models with frailty term failed to converge.

New comment 3. Please include description of the inclusion of taxa as time varying covariates in the methods.

New answer from authors: We have addressed this comment and added text in the methods section under the Survival Analysis Heading that reads (line 397) (new text is in red):

“Survival analysis. Time to delivery in the presence of specific bacterial taxa was visualized with Kaplan-Meier curves generated using the survfit() routine from the survival R package and hazard ratios with the corresponding p-values estimated using Cox proportional hazard regression models using coxph() routine of the survival R package. In these models, bacterial taxa were included as time varying covariates.”

Minor comment 4.

Original comment 4. The decrease in the total bacterial burden is an interesting finding. Is this decrease observed over time within women or could this be an artefact of missing visit 3 samples in women who had a PTB?

Answer from authors: The following figure shows a scatter plot of log₁₀ bacterial absolute abundance values vs gestation age (GA). As we can see there is a clear pull towards low bacterial absolute abundances among sPTB subjects at GA weeks 25,26,27. The table below the figure shows p- values of t-test comparing log₁₀ bacterial absolute abundances between TERM and sPTB subjects within a specified GA week. Non-significance of the test at week 27 is most likely due to smaller number of sPTB samples than in previous weeks. Thus, the trend of lower bacterial absolute abundances among sPTB subjects is driven by a subpopulation of sPTB except at the 27 weeks of GA in this dataset.

New comment 4. I apologize if this original comment was unclear. I meant to suggest that some additional analysis would help determine whether the difference is due to a decrease in load (as suggested by the heading

in the results) or whether those women with higher gestational ages in the sPTB group with low load are driving the difference. This can be determined by 1) re- doing the analysis above but only including those women with later GA samples and 2) looking at subject-specific trends over time within women in each group. If the subject specific plots indicate that the trajectories over time are relatively flat within each woman then difference observed at the later times is not due to an actual decrease over time. Alternatively, the authors could simply change the heading of this section to “Bacterial taxa AA is lower after 24 weeks...” to better reflect that the difference may not be due to a decrease over time.

New answer from authors: We have changed the heading of the section which now reads (line 1263): “Bacterial taxa absolute abundance is lower after 24 weeks in women who delivered preterm.”

Minor comment 5.

Original comment 5. Is a difference of 0.3 logs for beta-defensin-2 clinically meaningful? With a large sample size, small effect sizes can be statistically significant but may not be clinically different. This is also true of the individual taxa selected, the median relative abundance is very small.

Answer from authors: It is difficult to interpret the clinical significance of any difference until these biomarkers are validated in another clinical trial outside of the case-control study. Further, because this is the first report of microbial immune correlation, it is difficult to contextualize and interpret without a validation. In addition, Figure S6 suggests the difference might be important as these differences are higher between races.

New comment 5: If there is room, it would be helpful if points made in the author’s response could be added to the text.

New answer from authors: The following text was added to the manuscript (line 207-209): “However, the clinical significance of any differences or microbial immune correlations cannot be fully interpreted until these biomarkers are validated in another clinical trial outside of the case-control study.”

Minor comment 6.

Original comment 6. What is the reasoning behind removing samples with fewer than 1000 sequences, was the total bacterial burden low for these samples?

Answer from authors: This is a common practice in the field of microbiome analyses. These samples usually represent suboptimal processing. Low sequence count is in general a reflection of sequencing process random variation (including DNA quality, etc) and setting a threshold for sample read count reduces relative abundance inflation bias for highly abundant taxa due to non-detection of low abundance taxa.

New comment 6: In my experience, these samples are only removed if the total bacterial load is low, if there is sufficient load but low number of sequences the sample is re-sequenced. I wasn’t sure if this was the case in this study. I have no further suggestions on this point.

New Answer from authors: Thank you. No change were made.

Minor comment 9.

Original comment 9. How were the 0’s handled prior to the log transformation for the relative abundance?

Answer from authors: In all analyses of dependence on the frequency of sPTB as a function of log relative abundance of a given phylotype, 0’s were removed from the analysis as it does not make sense to include them where we ask how the frequency of sPTB depends on relative abundance of a given phylotype. We have performed analyses of dependence of sPTB frequencies on presence/absence of a given phylotype but have not seen any significant signals.

New comment 9: Although I think one could make a compelling argument why those samples with 0’s do carry information and shouldn’t be excluded, at the very least the exclusion of these samples should be clarified in the text so that the reader understands what was done and can evaluate results appropriately within this context.

New Answer from authors: We have addressed the reviewer comments and included new text to clarify the exclusion of samples where a given taxa was not detected (line 349-352):

“In this analysis, we only included samples in which a given bacterial taxon was detected, as no significant difference in the proportions of samples from the sPTB or term groups were found when only considering samples in which a given bacterial taxa was not detected.”

Errors

Original comment. Figure 1c&d (right) & figure S2, what do the n’s reported in the top of the plots refer to and why do they change for each taxa?

Answer from the authors: The n in the top of each plot represent the number of samples containing the phylotype analyzed and included in the analysis. The figure legends have been corrected and now reads:

New comment: Please revise the figure legends to explain the n displayed in the top of each plot.

Answer from the authors: Text to clarify what n or N means in each figure 1, 2, but also figure S1, S2 and S3.

Additional Suggestion:

1. “All scripts used in statistical analyses are available in the Supplemental data” – remains in statistical analysis section despite response to reviewer 1, please remove as reader is referred to the Github repository in the Methods. I very much appreciate the posting of the code and think this will be a useful reference for readers.

Answer from the authors: We also agree that the availability of the script is critical and could serve as reference to other scientists. We have modified the text and included that the code is available in GitHub. The text now reads (line: 326): All scripts used in statistical analyses are available in Github at <https://github.com/ravel-lab/M> and [M](https://github.com/ravel-lab/M)

2. From the code, it looks as though the mixed effect logistic regression was actually approximated with a Poisson distribution. Please clearly state this in the methods.

Answer from the authors: We have modified the text on line 327-329, which now reads (new text in red):

*“Mixed effects **Poisson** regression models were used to assess difference in the frequency of CSTs in women who delivered at term and those who delivered preterm, stratified by race or birth outcome (term vs sPTB).”*